# Estimation of cell lineages in tumors from spatial transcriptomics data

Beibei Ru [1,6], Jinlin Huang[2,3,6], Yu Zhang[1,2,5], Kenneth Aldape [4] &
Peng Jiang [1]✉

Spatial transcriptomics (ST) technology through in situ capturing has enabled topographical gene expression profiling of tumor tissues. However, each capturing spot may contain diverse immune and malignant cells, with different cell densities across tissue regions. Cell type deconvolution in tumor ST data remains challenging for existing methods designed to decompose general ST or bulk tumor data. We develop the Spatial Cellular Estimator for Tumors (SpaCET) to infer cell identities from tumor ST data. SpaCET first estimates cancer cell abundance by integrating a gene pattern dictionary of copy number alterations and expression changes in common malignancies. A constrained regression model then calibrates local cell densities and determines immune and stromal cell lineage fractions. SpaCET provides higher accuracy than existing methods based on simulation and real ST data with matched double-blind histopathology annotations as ground truth. Further, coupling cell fractions with ligand-receptor coexpression analysis, SpaCET reveals how intercellular interactions at the tumor-immune interface promote cancer progression.

Profiling the transcriptome of cells in their spatial context is critical to a mechanistic understanding of tumor progression and therapeutic resistance[1]. Recent years have seen the rapid development of spatial transcriptomics (ST) with gene coverage from a few targets to genome-wide and various cellular resolutions from subcellular to multiple cells[2,3]. As a key branch of ST methods, in situ capturing strategy based on positional molecular barcodes enables unbiased capture of the whole transcriptome within intact tissue[3]. Its representative techniques include Slide-seq[4], 10x Visium[5], and the early in situ capturing method from which Visium was developed[6]. Specifically, the commercial Visium platform can profile mRNA levels in freshfrozen and formalin-fixed paraffin-embedded (FFPE) tissues, enabling their widespread application[7]. However, the spatial spot of various capturing strategies with a 10–100 μm diameter might measure a mixture of signals from multiple cells of different lineages.

Consequently, decomposing cell identities in spots is a critical step in characterizing the spatial cellular landscape of tissues.

Many methods exist for cell type decomposition in general ST data[8–13] and bulk transcriptome profiling[14–17]. However, it is challenging for these methods and their underlying strategies to address the unique issue of tumor ST data. Several methods, such as Stereoscope[8], RCTD[11], and CIBERSORTx[15], predict cancer cell fractions relying on the availability of suitable malignant reference profiles. Other methods, such as EPIC[14], estimate malignant cell fraction without references by estimating the unknown cell fraction not covered by predefined cell signatures. However, this strategy does not distinguish malignant cells from truly unknown cell types[14]. Further, cellular density may vary significantly across tumor regions; thus, cell fractions decomposed by existing methods, which normalize overall fractions to 1 in each spot, are incomparable across different locations.

[1]Cancer Data Science Lab, Center for Cancer Research, National Cancer Institute, National Institutes of Health, Bethesda, MD, USA. [2]Department of Clinical Oncology, Li Ka Shing Faculty of Medicine, The University of Hong Kong, Hong Kong, China. [3]Department of Pathology, Sun Yat-sen University Cancer Center, Guangzhou, Guangdong, China. [4]Laboratory of Pathology, Center for Cancer Research, National Cancer Institute, National Institutes of Health, Bethesda, MD, USA. [5]Present address: Sun Yat-sen University Cancer Center, State Key Laboratory of Oncology in South China, Collaborative Innovation Center for Cancer Medicine, Guangzhou, China. [6]These authors contributed equally: Beibei Ru, Jinlin Huang. ✉e-mail: peng.jiang@nih.gov

Another strategy to deconvolve cell fractions is to generate the single-cell RNA sequencing (scRNA-seq) data paired with ST data on the same tumor sample[18,19]. However, single-cell experiments are challenging in frozen or FFPE tissue samples because it requires fresh samples and additional costs. Even if fresh samples are available, scRNA-seq may not reliably capture certain cell types, such as neutrophils, due to their sensitivity to rapid RNA damage[20].

This study introduces a computational framework, SpaCET (Spatial Cellular Estimator for Tumors), to decompose cell identities in tumor ST data. SpaCET addresses the challenges of tumor heterogeneities, tissue density variations, immune cell integrity, and collinearity among sublineages, which are not sufficiently considered in existing deconvolution methods. SpaCET outperforms other methods on eight tumor ST datasets spanning seven cancer types based on double-blind histopathology annotations. Moreover, SpaCET uncovered several potential cell−cell interactions supporting tumor progression. The source code for SpaCET is publicly available at https://github.com/data2intelligence/SpaCET.

## Results

### Decomposing cell lineages in tumor spatial transcriptomics

The SpaCET framework estimates cell lineages and intercellular interactions in tumor spatial transcriptomics data in three stages (Fig. 1a).

First, SpaCET estimates malignant cell fractions based on a gene pattern dictionary of copy number alterations (CNA) and malignant transcriptome signatures across common tumor types (Fig. 1b and Supplementary Fig. 1a). Most tumor ST datasets do not have matched scRNA-seq data as malignant cell reference. Alternatively, a consistent feature of most human tumors is chromosomal instability leading to common CNA patterns in each cancer[21]. Additionally, in chromosomal stable tumors with low CNA, malignant cells may still have transcriptome characteristics differentiating tumors from normal cells[22]. Thus, we created a gene pattern dictionary of CNA or tumor-normal expression differences from ~10,000 patient samples spanning 30 tumor types from the Cancer Genome Atlas (TCGA) (Supplementary Fig. 1b, c and Supplementary Table 1). In each tumor ST data, SpaCET searches for malignant cell spots whose expression profiles correlate with the CNA or expression pattern of the relevant tumor type (see "Methods").

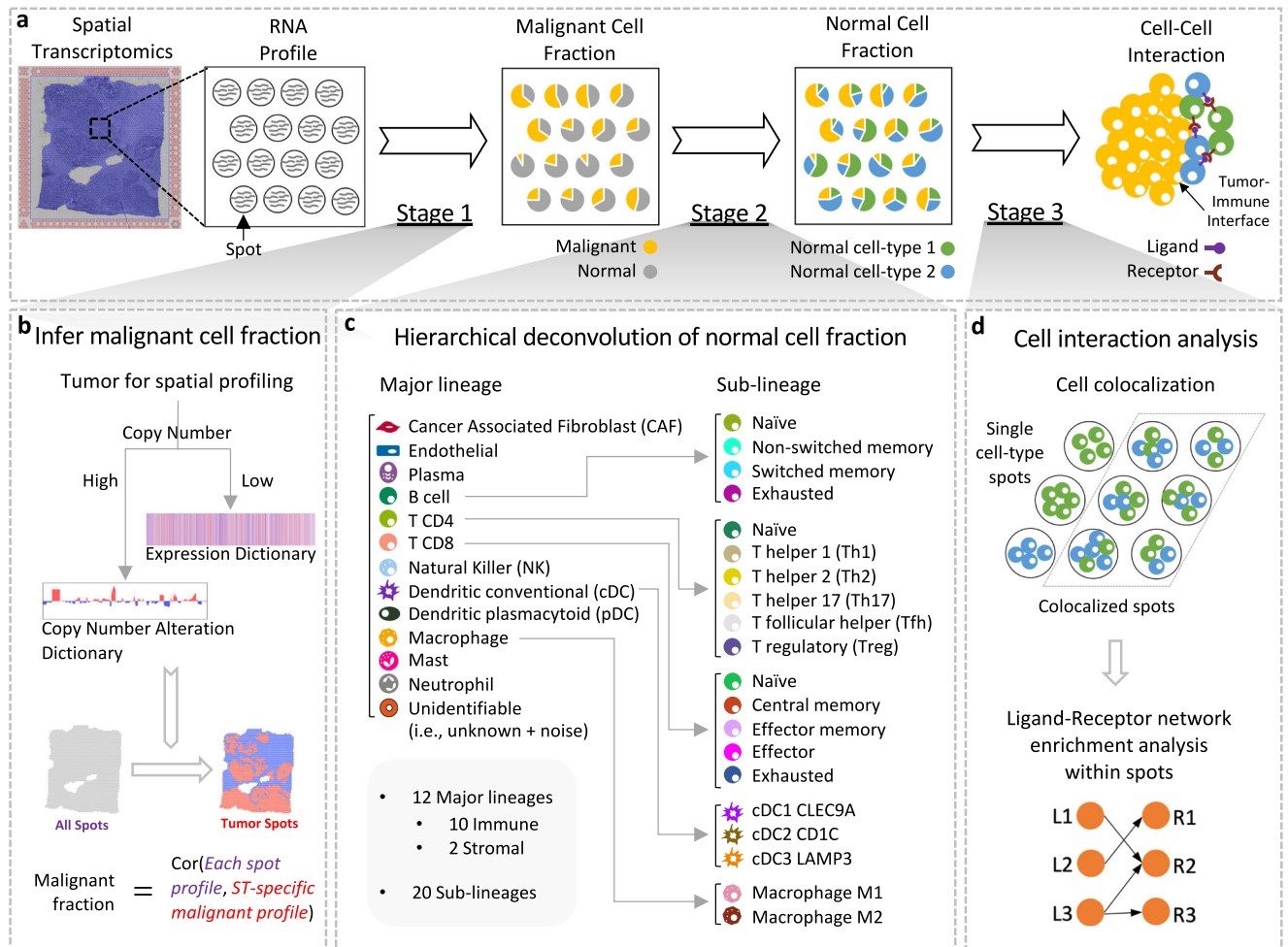

**Fig. 1 | Inferring cell fractions and interactions in tumor spatial transcriptomics. a** Three stages from input spatial transcriptomics (ST) data to cell lineage fractions and intercellular interactions. **b** Malignant cell fraction inference through a gene pattern dictionary. For a tumor ST dataset, SpaCET uses a dictionary of copy number alterations or tumor transcriptome patterns to identify tumor spots and further computes an ST-specific malignant expression profile. Then, SpaCET correlates the ST-specific malignant profile with the expression profile of each spot and normalizes the correlation coefficients to 0−1 as the malignant fractions of all spots. **c** Hierarchical deconvolution of nonmalignant cell fractions. Based on a hierarchical cell reference from the public scRNA-seq data atlas, SpaCET utilizes a constrained linear regression to estimate cell fractions on two levels. For level one, SpaCET decomposes the nonmalignant cell fractions into major lineages and unidentifiable components. For level two, major lineage fractions are further decomposed into corresponding sublineage fractions. **d** Cell−cell interaction analysis by testing cell colocalizations and ligand−receptor interactions. Based on inferred cell fractions, SpaCET measures cell colocalization through correlations across spots. Then, for the cell-type colocalized spots, SpaCET tests the significance of ligand−receptor co-expression as further evidence of physical interaction.

Second, SpaCET deconvolves nonmalignant cell fractions and adjusts cell densities under a unified linear model (Fig. 1c). Using scRNA-seq datasets from diverse cancer types, we defined reference expression profiles of immune and stromal cells in a hierarchical lineage (Fig. 1c and Supplementary Table 2). SpaCET utilizes a constrained linear regression to estimate cell lineages on two levels. SpaCET first decomposes nonmalignant cell fractions into immune lineages, stromal lineages, and unidentifiable components that our cell lineage reference cannot explain (Fig. 1c). The "unidentifiable" category enables our linear model to reduce the estimated cell fractions for ST spots with low cellular content or unknown cell types ("Methods"). SpaCET then further deconvolves immune sublineage fractions constrained on their parental lineage fractions (Fig. 1c). Expression signatures of closely related cell types can result in colinearity, leading to high result variations[23]. The hierarchical decomposition scheme will confine any result variances in sublineages due to collinearity from affecting cell fractions on the higher level.

Third, SpaCET infers intercellular interactions based on cell colocalization and ligand–receptor co-expression analysis (Fig. 1d and "Methods"). We focus on close contacts between cells within the same ST spot rather than between different ST spots because the gap between spots (e.g., -50 μm for Visium platform) may span several cells[24]. Linear correlations of cell fractions are computed across all ST spots to evaluate cell-type colocalization. High positive correlations indicate that cell-type pairs tend to colocalize together. To infer physical interactions, we further test the co-expression of ligand and receptor genes within the same ST spot for the colocalized cell-type pairs.

## Performance validation by simulated ST data

To evaluate the performance of SpaCET, we deconvoluted the simulated (this section) and real ST data with double-blind histopathology annotations (next section). The simulated ST dataset was generated by mixing 3–10 single-cell transcriptomic profiles from one scRNA-seq dataset to imitate the signal of a spot. Thus, the actual fraction of cell types at each synthetic ST spot was known. As evaluation metrics, we calculated Pearson correlation ($r$) and root-mean-square error (RMSE) between the decomposed cell fractions and the actual cell mixing ratios in simulations.

We collected 10 scRNA-seq datasets from melanoma[25,26], breast[27], colorectal[28], head and neck[29], liver[30], and non-small cell lung[31,32] cancers (Supplementary Table 2). Each study included thousands of single cells from various cell lineages, which allowed us to build a comprehensive cell atlas in tumor microenvironments (Fig. 2a, b and Supplementary Fig. 2a). We conducted intra- and inter-dataset validation to evaluate the decomposing performance on synthetic ST data. For the intra-dataset validation, each scRNA-seq dataset was split into two groups equally by patients. One patient group was used to generate cell reference profiles, and other nonoverlapping patients were used to generate synthetic ST data. For the inter-dataset validation, reference profiles and synthetic ST data were built from distinct scRNA-seq datasets.

Most intra-dataset validations of cell-type decomposition achieved high accuracies (Fig. 2c and Supplementary Fig. 2b, right boxplot). The performance is also robust if the read counts of simulated ST data stay within 50% of the single-cell reference data (Supplementary Fig. 2c). Additionally, specific lineages, such as malignant cells, cancer-associated fibroblasts (CAFs), neutrophil cells, B cells, and plasma cells, have high prediction accuracies (Fig. 2c, top boxplot). However, closely related cell types were challenging to deconvolve, as sublineages have lower performance metrics than their parental lineages (Fig. 2c).

In the inter-dataset validation, the reference profile generated in one single-cell cohort can generally predict the cell fractions in ST data synthesized from other single-cell cohorts (Fig. 2d and Supplementary Fig. 2d, top boxplot). We also generated leave-one-out signatures for

the inter-dataset evaluation, where we created reference profiles using all datasets except the one left out to synthesize the testing ST data. On average, leave-one-out signatures outperformed individual profiles from one scRNA-seq dataset, indicating that integrating multiple scRNA-seq datasets can create a generalizable reference for cell decompositions in many tumor types (Fig. 2d). Moreover, SpaCET with a hierarchical regression outperformed the one decomposing all sublineages in one single deconvolution level (Supplementary Fig. 2e). Therefore, in the following analyses, we used a combined hierarchical reference by averaging transcriptomics profiles for each cell type from 10 scRNA-seq datasets ("Methods").

We also compared SpaCET with several representative ST and bulk cell-type deconvolution methods[8–17] for decomposing the synthetic ST data. SpaCET outperformed other methods in both major and sublineages (Fig. 2e and Supplementary Fig. 3) in the simulated setting. In addition, we computed the running time and memory consumption of all methods, and SpaCET is among the high-effective algorithm groups (Fig. 2f).

## Validation by real ST data with double-blind histopathology annotations

We next evaluated the performance of SpaCET with real tumor ST data. The generation of an ST profile typically provides a hematoxylin and eosin (H&E)-stained image from the same tissue slide. According to the H&E morphology, pathologists labeled the local tissue density and regions of tumor, stroma, lymphocyte, and macrophage without knowing any deconvolution results. We applied SpaCET to decompose eight tumor ST datasets[5,18,19,33] spanning seven tumor types (Fig. 3a and Supplementary Table 3). The consistency between double-blind pathology annotations and estimated cell fractions would indicate cell-type decomposition accuracy.

For example, one breast cancer ST dataset from 10x Visium[5] measured 22,953 genes on 3183 spots (Supplementary Fig. 4a). The H&E staining of the same ST tissue slide enabled pathology annotations of the tumor, lymphocyte, and stroma regions through cell morphology, and macrophages through hemosiderin deposition[34] (Fig. 3b). We observed that the breast cancer-specific CNA signature in SpaCET was activated to estimate the malignant cell fraction, indicating that this breast tumor is chromosomally unstable (Supplementary Fig. 4b). The decomposition results from SpaCET showed that the breast tumor comprises mainly malignant cells, cancer-associated fibroblasts (CAFs), endothelial cells, macrophages, and T CD4 cells (Supplementary Fig. 4c).

We observed that low-density regions have large unidentifiable components (Fig. 3c and Supplementary Fig. 4c, d). A possible explanation is that sparse tissue regions have a dropout phenomenon, where many genes have zero read counts in ST data[35]. Such dropout will bring high noise to reduce regression coefficients for known cell types[36]. Indeed, dropout rates across ST spots are proportional to the tissue density annotated by pathologists (Fig. 3c and Supplementary Fig. 4e).

As a simplification, SpaCET only provides one malignant cell type per tumor ST dataset. However, malignant cells in different spatial regions may present distinct cellular states determined by cancer evolutions and interactions with local environments (Supplementary Fig. 4f–h). Thus, SpaCET also provides additional steps to identify the substates of cancer cells.

Based on the H&E staining annotations, we compared the prediction accuracy for SpaCET and previous methods[8–17] by using receiver operating characteristic (ROC) curves (Fig. 3d, e). Although SpaCET does not require a malignant cell signature and has a built-in normal cell reference, most previous methods require a single-cell reference for deconvolution. Since most tumor ST data do not have matched scRNA-seq data from the same sample, we build a pan-cancer single-cell reference from our scRNA-seq data collection for existing tools

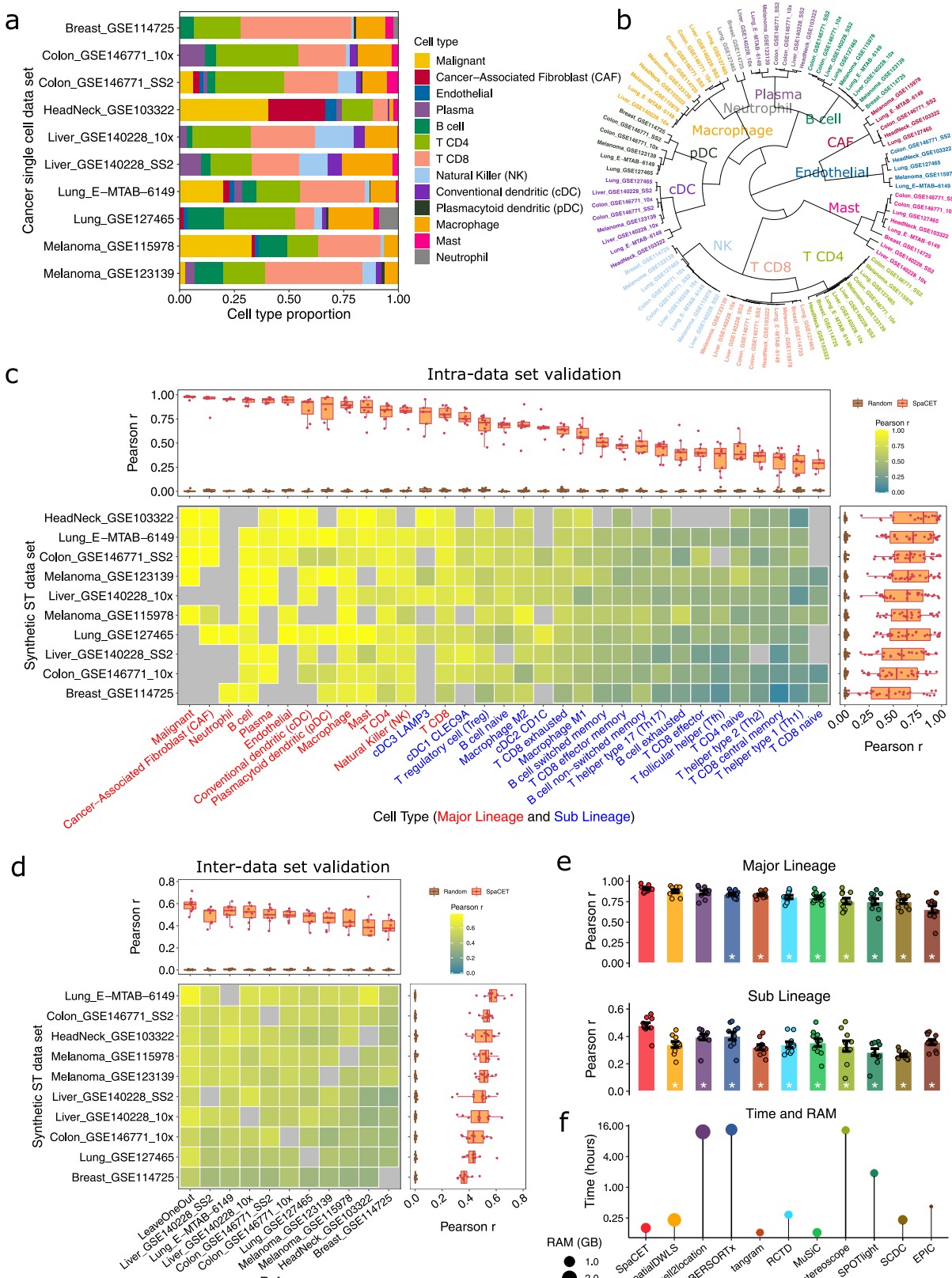

("Methods"). We found that SpaCET outperformed other methods for estimating malignant cells, stromal cells, and macrophages (Fig. 3e). On lymphocytes, several methods (e.g., Stereoscope and RCTD) also achieved high performance comparable to SpaCET.

We further evaluated seven more ST datasets on seven cancer types (Fig. 3f and Supplementary Figs. 5–11). In general, SpaCET

yielded more accurate estimates across cell types than other methods (Fig. 3f and Supplementary Fig. 12). SpatialDWLS and RCTD also achieved high performance for all cell types. Besides comparing SpaCET with other approaches, we also evaluated the robustness and effectiveness of SpaCET algorithm designs. By downsampling the tumor ST data from 4000 to 500 genes per spot, we found that

**Fig. 2 | Performance evaluation based on simulated ST data. a** Cell lineage proportions in 10 tumor scRNA-seq datasets used for ST simulation. **b** Hierarchical clustering of cell lineage reference profiles from scRNA-seq datasets based on marker gene set similarities. **c** Performance in intra-dataset validation for each scRNA-seq dataset (row) and cell type (column). The color in the heatmap presents Pearson correlation (r) between predicted versus known cell fractions. The gray color in the heatmap indicates missing cell types in the scRNA-seq dataset. Boxplots on the top present r values of the same cell type across all datasets (n = 10). Boxplots on the right present r values of all cell type predictions in the same dataset. We shuffled spot identities of cell type fraction vectors within each synthetic ST data and computed r values as random controls. For boxplots, the thick line represents the median value. The bottom and top of the boxes are the 25th and 75th percentiles (interquartile range). The whiskers encompass 1.5 times the interquartile range. **d** Performance of inter-dataset validation between scRNA-seq cohorts. The column and row labels show the scRNA-seq datasets (n = 10) used to generate cell-type reference profiles and synthetic ST data, respectively. The color in the heatmap presents the median Pearson correlation (r) between predicted and known cell fractions across all cell types. Boxplots and random controls are plotted as panel **c**. **e** Performance comparison between SpaCET and previous methods (color ordered in panel **f**). A dot represents a simulated ST dataset synthesized from a single scRNA-seq dataset (n = 10). The y value of an ST dataset presents the median Pearson correlation r between predicted and known cell fractions across cell types. All tools used the leave-one-out signature in panel **d**. The difference between SpaCET and other tools was evaluated by the two-sided Wilcoxon signed-rank test. A star indicates that SpaCET is significantly better than others (BH-adjusted p value <0.05). Bar height denotes the average value across simulated ST datasets; error bars denote standard errors. **f** Comparison of running time and memory consumption, using a simulated ST dataset of 1200 spots with default parameters.

SpaCET still keeps high performance on low-quality ST data (Supplementary Fig. 13a).

For the malignant cell quantification (Fig. 1b and Supplementary Fig. 1a), SpaCET prepared a pattern dictionary of both CNA and tumor-normal differential expression for diverse tumor types. All tumor ST data in our collection utilized the CNA pattern for cancer cell quantification due to significant correlations between spatial transcriptomic profiles and cancer type-specific CNA patterns (Supplementary Figs. 4–11). However, we still used these ST data as surrogates to evaluate the expression signatures prepared for chromosomal stable tumors. The expression signatures achieved comparable performance (Supplementary Fig. 13b), supporting the reliability of our expression-based procedure, which will start its role for CNA-low tumors. The unidentifiable component in the SpaCET regression model only brought minor performance improvements in deconvolving stromal cells (Supplementary Fig. 13c).

## SpaCET can decompose ST data with various resolutions
Different ST platforms have various resolutions of spatial capturing spots ranging from 10 to 100 μm. Thus, we evaluated whether SpaCET is applicable to a broad set of in situ capturing data with higher and lower resolutions. In our collected datasets (Supplementary Table 3), six of eight came from the 10x Visium with a detection spot diameter of 55 μm. For evaluations in data with a higher spatial resolution, we applied SpaCET to analyze a colon cancer Slide-RNA-seq dataset[33], consisting of 18,288 beads with a diameter of 10 μm covering 16,270 genes (Supplementary Fig. 10a, b). Pathologists annotated the matched H&E image for tumor and stroma regions based on cell morphologies (Fig. 4a). The deconvolution results show that SpaCET consistently outperformed other methods (Fig. 4b, c and Supplementary Fig. 10d). Since the diameter of a bead is 10 μm covering 1–2 cells, SpaCET may enable cell-type maps with a single-cell resolution (Fig. 4d, e and Supplementary Fig. 10e).

For evaluations in low-resolution data, we collected a pancreatic ductal adenocarcinoma dataset[19] generated by the early in situ capturing method (from which Visium was developed) with a spot diameter of 100 μm (Supplementary Fig. 11). SpaCET achieved a reliable performance compared to other approaches (last row in Supplementary Fig. 12). According to the matched scRNA-seq data[19], the current pancreatic tumor included acinar and ductal cells, not included in the in-house cell-type reference of SpaCET (Fig. 1c). SpaCET can accept a matched scRNA-seq dataset as customized cell-type references (Supplementary Fig. 14a). SpaCET's decomposition for acinar and ductal cells agreed well with their annotated locations in the H&E image (Supplementary Fig. 14b, c).

## SpaCET reveals intercellular interactions in spatial contexts
Cell–cell interactions in tumors play a pivotal role in cancer progression and therapeutic resistance[37]. The scRNA-seq data may reveal intercellular communications via analyzing the ligand–receptor co-expression across cell types[38]. However, such analysis loses proximity information and thus may report false-positive interactions that never come into contact in space. The deconvolution results by SpaCET should enable cellular interaction analysis in spatial contexts.

To identify intercellular interactions from ST data, SpaCET utilized a two-step approach, assessing cell colocalizations followed by ligand–receptor co-expression analysis. Since the gap between ST spots may span multiple cells[24], our method investigated cell contacts within the same spot rather than between different spots. We calculated the Spearman correlation between cell-type pairs across ST spots based on estimated cell fractions. The strong positive correlation of a cell-type pair indicates their cell colocalization. For example, in a breast tumor, we identified several potential colocalized cell-type pairs, such as CAFs with endothelial cells and M2 macrophages (Fig. 5a and Supplementary Fig. 15a).

To rule out a high cell fraction correlation caused by similar reference profiles, we compared the correlations between cell-type fractions and between cell-type reference profiles. Although the correlation of CAF and endothelial cell fractions is high, their profile similarity is also proportionally high (Fig. 5b). However, the similarity between CAF and M2 macrophage references was relatively low (Fig. 5b), indicating that the CAF–M2 colocalization is not simply due to profile similarity.

## Ligand–receptor interactions within ST spots
Cell colocalization does not directly indicate physical interaction. Thus, we sought further evidence for cell–cell interactions by analyzing ligand–receptor (L–R) interactions within ST spots. From a previous study[39], we obtained approximately 2500 L–R pairs. We computed an L–R network score for each spot as the sum of expression value multiplications between L–R gene pairs, normalized by the average score from 1000 random L–R networks with the same connection degrees as the real network ("Methods").

The L–R network score at each ST spot indicates the intensity of ligand–receptor interactions at each location (Fig. 5c and Supplementary Fig. 15b), but not specific interactions between cell types. Thus, SpaCET further performed an enrichment analysis of L–R network scores for each cell-type pair. For example, for the colocalization between CAF and M2 cells in the breast tumor tissue, SpaCET grouped all ST spots into four categories: CAF–M2 colocalized, CAF or M2 dominated, and others (Fig. 5d, e). We found that CAF–M2 colocalized spots have more substantial L–R network scores than CAF/M2-dominated spots (Fig. 5f). In contrast, there was no significant difference between CAF-endothelial colocalized and CAF/endothelial-dominated spots (Supplementary Fig. 15c, d). These results lead to the prediction of CAF–M2 interactions but not CAF-endothelial interactions. Meanwhile, the CAF–M2 interaction is consistently significant using estimated cell-type fractions from different deconvolution methods (Supplementary Table 4).

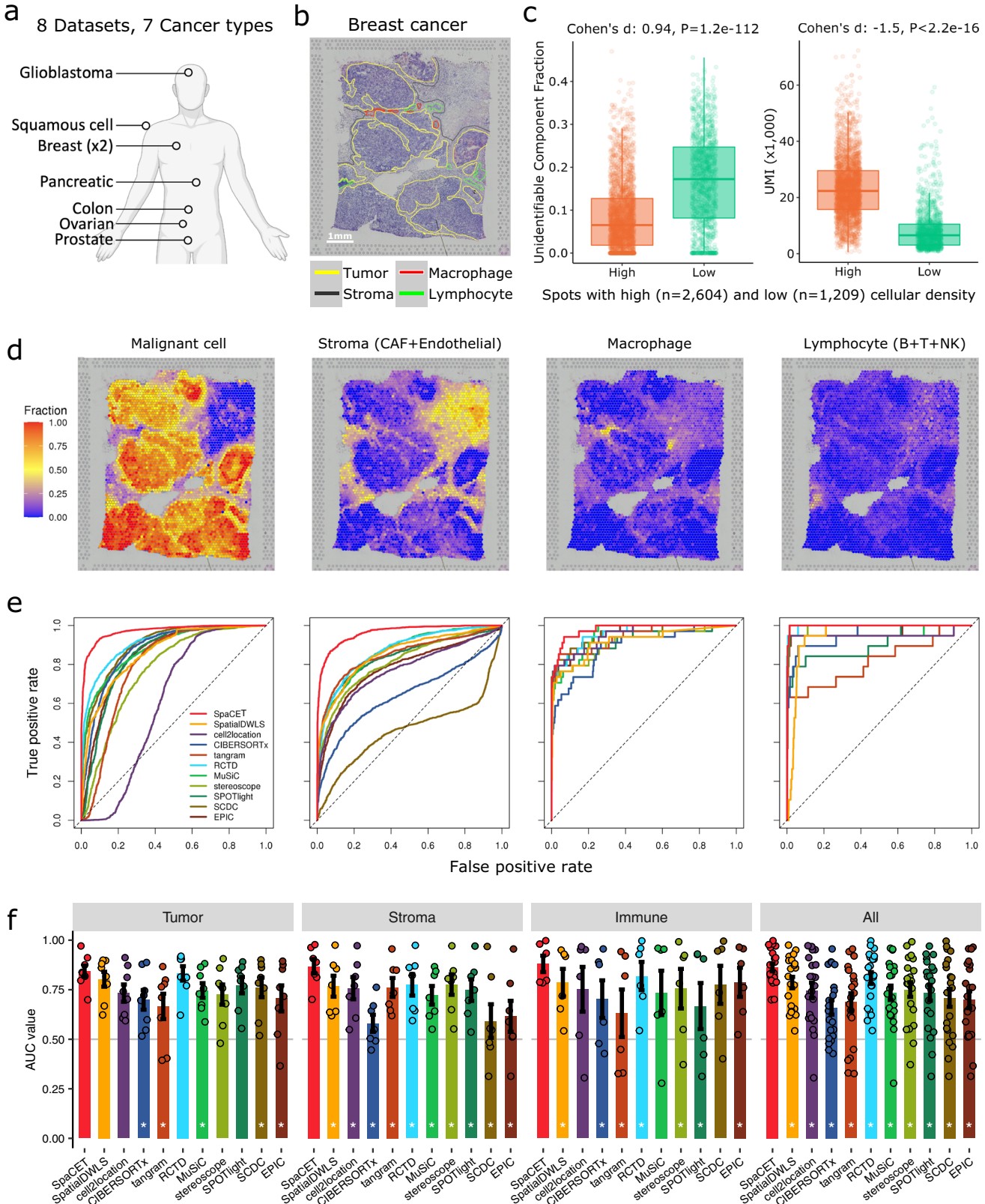

By integrating our collected scRNA-seq datasets (Supplementary Table 2), we also identified several putative L–R pairs mediating the crosstalk between CAFs and M2 macrophages for the current breast tumor (Fig. 5g and "Methods"). The advantage of integrating single-cell and spatial data is that we could exclude two categories of false positives: (1) L–R pairs with spatial proximity but not from different cell types, thus they do not represent interactions between distinct cell types; and (2) L–R pairs identified from single-cell data but without spatial proximity, thus they do not represent physical contacts between cells (Supplementary Fig. 16 and "Methods").

## Cell–cell interactions at the tumor-immune interface

The SpaCET results enabled the systematic analysis of biological functions of cell–cell interactions in a spatial context. For example, the

**Fig. 3 | Performance validation based on double-blind pathology annotations. a** Multiple tumor ST datasets used for performance evaluation. The human body outline was generated using BioRender. **b** An example hematoxylin and eosin (H&E) image with double-blind pathology annotations. **c** Unidentifiable component fractions (left) and unique molecular identifier (UMI) counts (right) across spots in both high and low cellular density regions. The group values were compared by calculating the Cohen's $d$ effect size and the two-sided Wilcoxon rank-sum test. For the boxplot, the thick line represents the median value. The bottom and top of the boxes are the 25th and 75th percentiles (interquartile range). The whiskers encompass 1.5 times the interquartile range. **d** Fractions of malignant, stromal, macrophage, and lymphocyte cells, decomposed by SpaCET. **e** Receiver operating characteristic (ROC) curves of cell fraction prediction. This example is based on the cell region annotation in panel **b**. For each method, the ROC curve presents false-positive rates against true-positive rates at different thresholds of cell fraction across spots. **f** Performance comparison among methods. Each dot represents a dataset ($n = 8$ for each bar). $y$-axis presents the area under the ROC curve (AUC) value of cell fraction decompositions for each method. The subpanels represent the results in distinct tumor regions, and the last subpanel considered data from all three region types together. In each subpanel, the difference between SpaCET and other tools was evaluated by the two-sided Wilcoxon signed-rank test. A star indicates that SpaCET is significantly better than others (BH-adjusted $p$ value <0.05). Bar height denotes the average value across ST datasets; error bars denote standard errors.

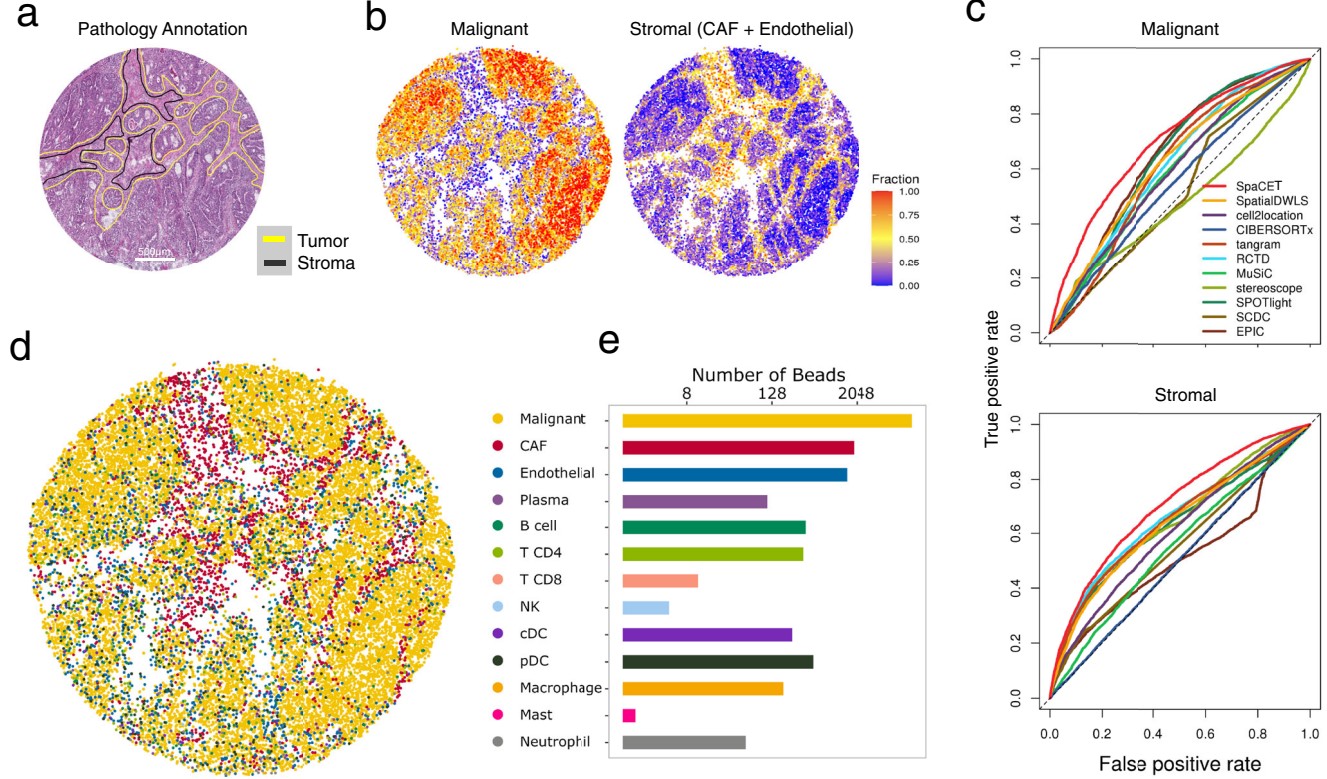

**Fig. 4 | Application of SpaCET to a colon cancer Slide-seq dataset. a** The H&E-stained image with double-blind pathology annotations. **b** Fractions of malignant and stromal cells, decomposed by SpaCET. **c** ROC curves of cell fraction prediction based on the annotation in panel **a**, shown as Fig. 3e. **d** Spatial localization of cell major lineages. The cell type of a bead is defined by the most abundant cell type in this bead. **e** Number of beads for each cell type.

spots of CAF−M2 interactions identified in the breast tumor example are significantly close to the tumor-immune interface (Fig. 6a, b). We, therefore, classified malignant cells as "close" or "distant", based on their distance from ST spots with CAF−M2 interactions (Fig. 6c). We generated differential gene expression profiles between close and distant spots followed by gene set enrichment analysis (GSEA) (Supplementary Fig. 17a). Distant malignant cells showed enrichment in cell-cycle pathways, a result typical for fast-growing cancer cells (Fig. 6d). In contrast, genes upregulated in the close malignant cells primarily belonged to epithelial-mesenchymal transition (EMT) pathway (Fig. 6d and Supplementary Fig. 17b).

Additionally, GSEA reported several genes from the EMT pathway (e.g., *COL1A1*, *LRRC15*, and *LUM*), which are highly expressed in the malignant cells close to CAF−M2 interaction regions (Fig. 6e). These genes have been demonstrated to drive cancer progression in breast cancer and other cancer types. For example, collagen type I alpha 1 (*COL1A1*) is abundant in the extracellular matrix of breast cancer cells, and knockdown of *COL1A1* in cancer cell lines inhibits cancer cell migrations[40]. By upregulating MAPK signaling, lumican (*LUM*)

promotes the proliferation and migration of bladder cancer cells[41]. In-vitro and in-vivo studies show that overexpression of leucine-rich repeat containing 15 (*LRRC15*) augments metastasis in multiple cancer types (e.g., breast cancer, osteosarcoma, and soft tissue sarcomas)[42]. However, additional experimental evidence is still needed to validate the biological functions of these genes in promoting cancer aggression.

## Discussion

We present SpaCET with several algorithmic designs to address the challenge of decomposing cell fractions in tumor ST data. We demonstrated its superior performance over existing deconvolution methods on ST data with a broad range of cellular resolutions. SpaCET estimates cancer cell fractions based on a gene pattern dictionary of copy number alterations and expression changes across tumor types, which performs better than the inferCNV-based strategy[43] in both prediction accuracy and running efficiency (Supplementary Fig. 13d). This dictionary-based approach should also be broadly applicable to malignant cell identification in scRNA-seq data analysis, particularly for

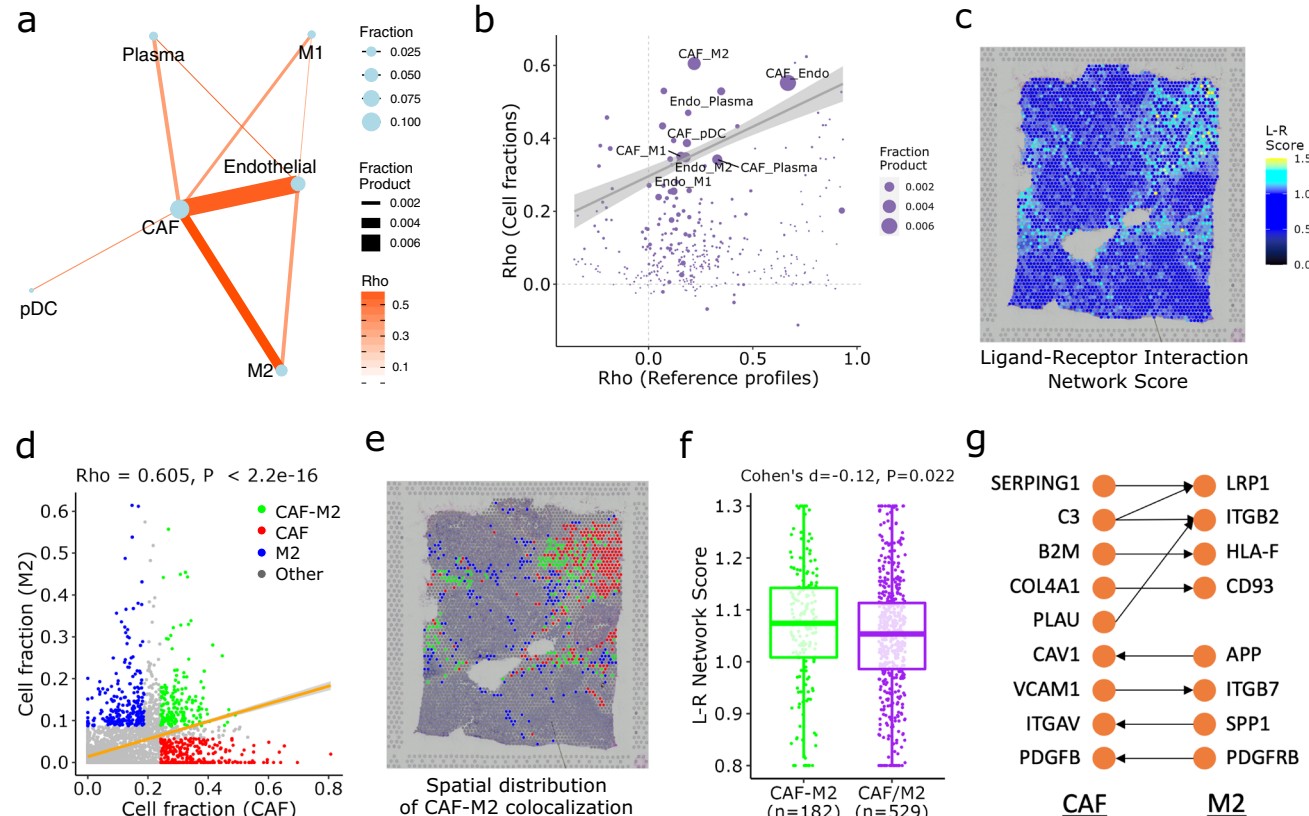

**Fig. 5 | SpaCET identifies intercellular interactions in the breast tumor.**
**a** Spearman correlations of cell-type fractions across breast tumor ST spots. Each node in the network represents a cell type, and the size of a node refers to the average fraction of this cell type across all spots. Each edge represents the colocalization of a cell-type pair, and the size of an edge refers to the fraction product of this cell-type pair. **b** Spearman correlation analysis between reference profiles (*x*-axis) and between cell-type fractions (*y*-axis). Each dot represents a cell-type pair. The straight line presents the weighted linear regression result with the gray shadow as the 95% confidence interval. **c** Ligand−receptor interaction network scores for all spots. **d** CAF and M2 fractions across all spots. Each dot represents an ST spot. According to the CAF or M2 cell fractions, spots were grouped into four categories: CAF−M2 colocalized (top 15% in both CAF and M2, *n* = 182), CAF-dominated (Top 15% in CAF and bottom 75% in M2, *n* = 295), M2 dominated (Top

15% in M2 and bottom 75% in CAF, *n* = 234), and others (*n* = 3102). The straight line presents the linear regression between the cell fractions of CAF and M2 with the gray shadow as the 95% confidence interval. The Rho and *p* values are computed from the two-sided Spearman correlation test (*n* = 3813 spots). **e** Spatial distribution of CAF-M2 colocalized and CAF/M2-dominated spots in panel **d**. **f** Difference of L−R interaction network score between CAF-M2 colocalized spots and CAF/M2-dominated spots in panel **d**. For the boxplot, the thick line represents the median value. The bottom and top of the boxes are the 25th and 75th percentiles (interquartile range). The whiskers encompass 1.5 times the interquartile range. Group values were compared by Cohen's *d* effect size and two-sided Wilcoxon rank-sum test. **g** L−R pairs mediating the CAF-M2 interaction in the current breast cancer tissue. The direction of an arrow source from ligand to receptor.

chromosomal stable tumors where the conventional inferCNV package does not work.

Our strategy of hierarchical decomposition constrains the negative effect of collinearity among closely related cell types within a sublineage of decomposition results (Supplementary Fig. 2e). A previous tool, MuSiC[16], also performs hierarchical deconvolution by asking users to define cell hierarchies manually based on the clustering results of scRNA-Seq data. As an advantage, SpaCET provides a comprehensive hierarchical cell-type reference summarized from many tumor types. Moreover, SpaCET includes an unidentifiable component to address cell types missing from the reference and cell density variations across regions (Supplementary Fig. 4d, e and Supplementary Fig. 13c).

The spatial heterogeneity in different tumor regions is driven by tumor evolution and intercellular interactions between cancer cells and immune/stromal cells[44]. Although several strategies[45–47] have been developed for exploring cell−cell communications in subcellular resolution ST technologies (e.g., seqFISH+ or CODEX), they are not applicable to multiple cellular (i.e., spot level) ST data because of gene coverage and resolution differences. SpaCET can combine colocalization and ligand−receptor analysis to study cell−cell interactions in multiple cellular ST data.

Several limitations exist for SpaCET. First, when estimating the malignant cell fractions (Fig. 1b), the ST-specific malignant expression profile was generated by averaging the expression profiles of all identified malignant cell spots within tumor ST data. As such, the estimation accuracy of malignant cell fraction might decrease for tumor ST data containing very distinct cancer cell states. Second, during the deconvolution process, we assume that the marker genes of reference cell types do not express in unknown cell types. However, our simulation analysis showed that high expression of more than 30% of existing reference marker genes in unknown cell types would lead to underestimated unknown cell fractions (Supplementary Fig. 18). If this happens, there should be two fixes: (1) the unknown cell type may come from close lineage to existing cell types, thus should be classified as known cell types instead of unknown ones; (2) the users should pre-estimate existing cell lineages for new tissue. If SpaCET's default lineages do not comprehensively cover the cell repertoire in the input sample, users might need additional cell signatures and run SpaCET with customized references.

SpaCET is a framework for understanding the spatial organization of cells in tumors and how spatial organizations influence cancer progression. With the continuous accumulation of spatial transcriptomics data from clinical studies, we foresee that SpaCET will

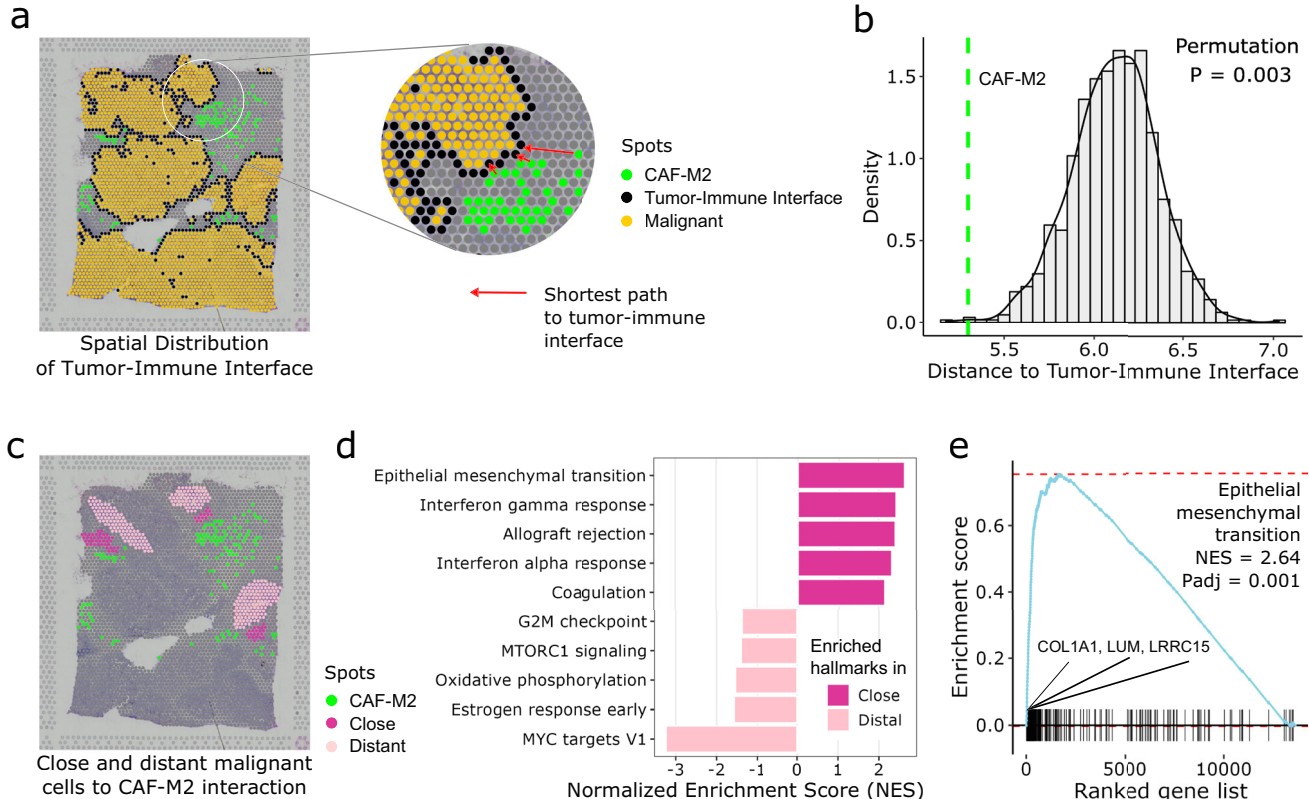

**Fig. 6 | Association between CAF–M2 interactions and malignant cell invasion. a** The CAF–M2 spots at the interface between the tumor and immune regions in a breast tumor example. The distance between a CAF–M2 spot and a tumor-immune interface is the shortest path of this spot to the interface. **b** Distance between CAF–M2 interaction spots and tumor-immune boundaries. The green line represents the average distance between each CAF–M2 spot and the tumor-immune interface. The null distribution of distance was computed through 1000 randomizations. **c** The spots of close and distant malignant cells relative to CAF–M2 interaction spots. **d** Gene set enrichment analysis (GSEA) of the differential expression between close and distant malignant cells to CAF–M2 spots conditioned on malignant cell fractions. **e** The GSEA enrichment plot of epithelial-mesenchymal transition pathway from panel **d**. The x-axis represents the gene list ranked by the differential expression analyzed in panel **d**. The black vertical bars along the x-axis represent genes from the pathway labeled. If all genes (vertical bars) of a pathway tend to be enriched in the left-most part of the x-axis, it indicates that this pathway is active in close malignant cells, and vice versa. The cyan line is the enrichment curve of the pathway, and the red dashed line refers to the maximum and minimum value of the cyan line, respectively. The p value is computed through the two-sided permutation test (n = 1000 randomizations) adjusted by the Benjamini–Hochberg procedure.

provide mechanistic insights underlying many oncogenic processes and therapeutic solutions to bottlenecks of current antitumor treatments.

## Methods
### SpaCET framework design
SpaCET consists of three sequential stages to estimate the cell lineages and intercellular interactions from tumor spatial transcriptomics (ST) data.

**Stage 1. Malignant cell fraction inference.** We designed a three-step process (Supplementary Fig. 1a) to infer malignant cell fractions without any reference based on a dictionary of cancer type-specific gene patterns (Supplementary Table 1 and see the following section: "Build a dictionary of cancer type-specific signatures").

(1) *Clustering all spots from a tumor ST dataset.* All spots from a tumor ST dataset were clustered by hierarchical clustering, and then the hierarchical tree was cut into K clusters by specifying K = 2−9. The optimal number of clusters was determined as the point preceding the largest decrease in the silhouette value, which measures how similar an object is to its own cluster compared to other clusters.

(2) *Determining the malignant cell clusters.* The expression profiles of all spots within a tumor ST dataset were firstly mean-centralized across all spots for each gene. Then, the expression profile of each

spot was correlated to the cancer type-specific (CNA or expression) signature at a genome-wide scale with Pearson correlation. The spot clusters were identified as malignant cell clusters by using the following two criteria: (1) the average coefficient r values of the spots within a cluster is significantly greater than 0 (one-sided Wilcoxon signed-rank test, p < 0.05); (2) the proportion of spots positively correlated to the cancer type-specific signature (Pearson's r > 0 and two-sided correlation test p < 0.05) within a cluster is more than the proportion in the whole ST dataset.

When correlating the expression profiles of ST spots to the cancer type-specific signature, we set the cancer type-specific CNA signature as the first option because chromosomal instability is a consistent feature of human tumors[21]. Alternatively, if no spots significantly correlate with the CNA signature, the cancer type-specific expression signature would be activated. This situation might result from chromosomally stable malignant cells with low CNA. In the chromosomal stable tumors with low CNA, malignant cells may still have transcriptomics characteristics differentiating tumors from normal cells[22]. For the cancer types not included in our dictionary, we created a pan-cancer expression signature by averaging all TCGA cancer type-specific expression signatures.

(3) *Estimating malignant cell abundance across all spots.* For a tumor ST dataset, the ST-specific malignant expression profile was achieved by averaging the expression profiles of the spots in the

malignant cell clusters from step 2. Subsequently, the expression profile of each spot within a tumor ST dataset was correlated to the ST-specific malignant expression profile. The Pearson correlation coefficients ($r$) across all spots were normalized to 0–1 as estimated malignant cell fractions for a tumor ST dataset.

$$F_{\text{malignant}} = r/(r_{\text{top5\%}} - r_{\text{bottom5\%}}) \tag{1}$$

Where $r_{\text{top5\%}}$ and $r_{\text{bottom5\%}}$ are the average $r$ from the top and bottom 5% spots across spots sorted on $r$. Then, the nonmalignant ($F_{\text{nonmalignant}}$) cell faction of a spot is computed as

$$F_{\text{nonmalignant}} = 1 - F_{\text{malignant}} \tag{2}$$

**Stage 2. Hierarchical deconvolution of nonmalignant cell fractions.** Due to the high transcriptional similarity of cell sublineages (e.g., macrophage M1 and M2), collinearity will induce high result variance for regression analysis[23]. Thus, based on an expression profile tree of stromal and immune cells, we utilized a two-level hierarchical model to decompose the nonmalignant cell fraction. On level one, the fractions of major lineages are estimated; on level two, the sublineage fractions are generated based on their major lineage fractions.

Based on a hierarchical atlas of reference profiles in the tumor microenvironment derived from single-cell RNA-seq datasets from diverse cancer types (see the following section: "Reference profile of cell types in deconvolution"), a constrained linear regression model, non-negative least squares, is utilized to hierarchically estimate cell fractions in immune, stromal cell lineage, and unidentifiable components in transcripts per million (TPM) space. In the level one decomposition, we keep an unidentifiable component in case several spots include cell types or random noise (due to low cell density) that our collected cell reference cannot explain.

For a given spot, $S$ represents its expression profile with the same gene dimension as the cell reference profile used for deconvolution. The $R_{\text{malignant}}$ and $F_{\text{malignant}}$ donate the ST-specific malignant expression profile and the corresponding fraction in stage 1. Thus, the expression profile $S$ of nonmalignant cells for this spot can be computed as

$$S_{\text{nonmalignant}} = S - R_{\text{malignant}} \times F_{\text{malignant}} \tag{3}$$

The $R_c$ and $F_c$ donate the reference and fraction of nonmalignant cell type $c$. $c$ is the set of nonmalignant cell types on level one. $R_{\text{unknown}}$ and $F_{\text{unknown}}$ represent the reference and fraction of unknown cell types that our reference collection does not include. Given this, $S_{\text{nonmalignant}}$ can be written as

$$S_{\text{nonmalignant}} = \sum_{c \in C} R_c \times F_c + R_{\text{unknown}} \times F_{\text{unknown}} \tag{4}$$

Here, we assume that our computed marker genes (details in "Methods" section: "Reference profile of cell types in deconvolution") of immune and stromal cells were lowly expressed in unknown cell types. In other words, unknown cell types are sufficiently different from our existing immune and stromal cell types. Thus, $R_{\text{unknown}} \approx 0$ since all gene dimensions are within our marker gene set. The last equation can be simplified as

$$S_{\text{nonmalignant}} \approx \sum_{c \in C} R_c \times F_c \left(F_c \geq 0, \sum_{c \in C} F_c \leq F_{\text{nonmalignant}}\right) \tag{5}$$

This solution can be estimated by a constrained non-negative least squares optimization. Subsequently, the fraction of unidentifiable component is computed as

$$F_{\text{unidentifiable}} = 1 - F_{\text{malignant}} - \sum_{c \in C} F_c \tag{6}$$

$F_{\text{unidentifiable}}$ contains both unknown cell fraction and noise. The latter mainly results from the gene dropout within ST spots due to low tissue density (Supplementary Fig. 4d, e).

Based on the above results, the sublineage fractions are further estimated on level two. For example, the fraction of macrophages is divided into sublineages M1 and M2.

$$S_{\text{nonmalignant}} - \sum_{c \in C \& c \neq M\varphi} R_c \times F_c = R_{M1} \times F_{M1} + R_{M2} \times F_{M2} \tag{7}$$
$$(F_{M1}, F_{M2} \geq 0, F_{M1} + F_{M2} \leq F_{M\varphi})$$

**Stage 3. Cell–cell interaction analysis.** Based on cell lineage fractions inferred, SpaCET explores cell–cell interaction by analyzing both cell colocalization and ligand–receptor (L–R) co-expression analysis across all ST spots. The former demonstrates the co-occurrence of cell-type pairs, whereas the latter provides evidence for cell–cell physical contacts by sending and receiving signals.

Cell colocalization was evaluated using the Spearman correlation between cell-type pairs across all spots. To further evaluate the significance of cell colocalization, we calculated the Spearman correlation between their reference profiles to rule out high colocalization due to similar reference profiles. Previous methods, such as stereoscope[8], included the cell colocalization analyses, but without considering L–R interactions.

The overall level of L–R interactions within a spot was evaluated by 2558 L–R pairs collected from a previous study[39]. These ligands and receptors were filtered within genes detected by the ST platform. We shuffled the L–R interaction network by using BiRewire package[48] to generate 1000 randomized networks while preserving directed degree distributions. For a spot, an L–R network score is defined as the sum of expression products between all L–R pairs, divided by the average random value from 1000 randomized networks. $P$ values were calculated with the empirical null distribution generated from network scores of randomized L–R interactions.

$$\text{Network Score(NS)} = \frac{\sum_i E_{Li} \times E_{Ri}}{<\sum_i E_{Li} \times E_{Ri}>}, \ P \text{ value} = P_r(\text{NS}_{\text{random}} \geq \text{NS}) \tag{8}$$

$E_{Li}$ and $E_{Ri}$ donate the expression of ligand and receptor from the $i$th L–R pair, respectively. The <> represents averaging the product sums from 1000 random networks.

For a colocalized cell-type pair, SpaCET grouped all ST spots into four categories: cell-type pair colocalized, either single cell-type dominated, and others. This colocalized cell-type pair would be considered to have cell–cell communication and interaction in the current ST dataset if the colocalized spots have more substantial L–R network scores than the single cell-type dominated spots (two-sided Wilcoxon rank-sum test, $p < 0.05$).

Our strategy is distinct from CellPhoneDB[49] to interrogate L–R interaction for scRNA-seq analysis, which randomly permutes the cluster labels of all cells. This strategy does not apply to ST analysis because the expression profile of each cell in ST is unknown due to mixed transcriptomics signals within spots. Thus, the computed L–R network score at each ST spot from SpaCET indicates the overall intensity of L–R interactions at each spot, but not specific interactions between the two cell types.

## Build a dictionary of cancer type-specific signatures

Based on the SNP6 Array and RNA-seq data from The Cancer Genome Atlas (TCGA), we built a gene pattern dictionary of copy number alterations (CNA) and expression changes for 30 solid tumor types (Supplementary Table 1). The cancer type-specific CNA signature of a cancer type was computed by averaging bulk tumor CNA values on gene levels across patients. For each cancer type, the lower quartile of patients sorted by total CNA burdens was excluded before calculating

the CNA signature. The cancer type-specific expression signature of a cancer type was generated as log2 Fold Change of differential expression between tumor and normal samples by R package limma[50]. Several cancer types do not have expression signatures due to a lack of adequate normal samples ($n < 10$ patients). The pan-cancer expression signature was created by averaging all cancer type-specific expression signatures.

### Single-cell RNA-seq data collection
For the validation of SpaCET using ST data simulation and generation of cell-type references, we collected 10 single-cell RNA-seq datasets from diverse tumor types, including melanoma[25,26], breast[27], colorectal[28], head and neck[29], liver[30], and lung[31,32] cancers (Supplementary Table 2). These datasets cover various platforms, including 10x genomics, Smart-Seq2, and InDrop. The cell-type annotations were from original studies. We further split macrophages into M1 and M2 subtypes by using marker genes from a previous study[27]. We also used SingleR package[51] to split B, CD4 T, CD8 T, and cDC cells into their corresponding sublineages (Fig. 1c). The immune cell signatures presented comparable deconvolution performance compared to a recently published tumor-immune cell atlas[52] (Supplementary Fig. 13e).

### Reference profile of cell types in deconvolution
The reference profile of a cell type in a single scRNA-seq dataset was the average expression profile across all single cells for this cell type in non-log TPM space. The marker genes for a cell type were defined with the following two steps: (1) carrying out differential expression analysis through the R package limma[50] between this cell type and every other cell type in log2(TPM/10 + 1) space; (2) selecting genes (log2 (Fold Change) > 0.25 and adjusted $p$ value <0.01) from the top 500 overexpressed genes ranked by adjusted $p$ values.

The combined reference profile of a cell type was generated by averaging the reference profiles of this cell type in all scRNA-seq datasets collected in Supplementary Table 2. The markers of a cell type were the markers of this cell type that appeared in at least half of scRNA-seq datasets.

### Simulated spatial transcriptomics data
To evaluate our method SpaCET, we constructed simulated (synthetic) ST data based on scRNA-seq data. For both intra- and inter-dataset validation, each simulated ST data contains 1200 spots. The simulated ST data, derived from the scRNA dataset with malignant cells, has three different malignant cell fractions, i.e., 0% (500 spots), 50% (200 spots), and 100% (500 spots). For a spot, 1–3 cell types and 3–10 cells were randomly selected from the scRNA-seq dataset using Dirichlet distribution[8], a probability distribution of multiple cell type fractions that sums to 1. Subsequently, transcriptomic profiles of selected single cells were averaged as the synthetic mixture. Thus, the "ground truth" of the cell-type fraction at each spot was known.

### Spatial transcriptomics datasets
We collected eight tumor spatial transcriptomics datasets[5,18,19,33] on seven cancer types listed in Supplementary Table 3. The raw count matrices of ST data were normalized to TPM (equivalent to count-per-million CPM for 10x data) to be deconvolved by SpaCET. The exact matched H&E-stained tumor tissue images were annotated by pathologists for tumor, stroma, and lymphocyte regions through cell morphologies, and macrophage regions if hemosiderin deposition features are available[34].

### Comparison to alternative deconvolution methods
We compared SpaCET with several ST and bulk data decomposition approaches[8–17]. Each method was run with its default parameters on a machine with a 2.60 GHz 8-Core CPU with 32GM of RAM. Since SpaCET does not require a malignant reference and has a built-in normal cell

atlas (Fig. 1), we built a pan-cancer scRNA-seq reference for the other methods by using the 10 scRNA-seq datasets in this study (Supplementary Table 2). Fifty single cells were randomly chosen for each cell type. For the simulated ST data, both Pearson correlation $r$ and root-mean-square error (RMSE) were calculated between predicted cell fraction and ground truth. For the real ST data, area under the ROC curve (AUC) values were calculated for each annotated region from the H&E image by pathologists via double-blind annotations.

### Exploration of cancer cell states in spatial transcriptomics data
Based on the deconvolution results, the spots with highly abundant malignant cells (fraction > 0.7) were selected to carry out clustering analysis. The optimal number of clusters (i.e., cancer cell states) was determined by calculating the silhouette value. We then collected 16 cancer gene modules from a recent study[53], and computed the average expression level of each module for malignant cell spots to explore the function of two cancer cell states. The deconvolution process of cancer cell states was the same as estimating the immune sublineage (see "Methods": "SpaCET framework design"−stage 2).

### Identification of significant ligand−receptor pairs
To investigate L−R pairs mediating the CAF−M2 interaction in breast tumor tissue, we integrated both ST data and single-cell data because ST spot data contains the mixture transcriptome from a few cell types, thus cannot directly reveal the cell source of gene expression. First, we computed the Spearman correlation of ligand and receptor pairs across CAF−M2 colocalized spots. Then, we examined whether the highly correlated L−R pairs had significant co-expression between CAF and M2 across our collected scRNA-seq datasets (Supplementary Table 2) by using a similar strategy from CellPhoneDB[49]. Briefly, to identify significant L−R pairs from scRNA-seq data, we scored an L−R pair between cell-type X and Y as the product of average ligand expression across all cells in X and the average receptor expression across all cells in Y. The $p$ value was calculated based on a null distribution generated by shuffling the cell type of all cells within the X−Y pair (1000 times) and repeating the L−R interaction computation. The L−R analysis in this part aims to identify the specific L−R pairs mediating the CAF−M2 interaction, whereas the L−R analysis in stage 3 of SpaCET refers to the overall interaction level of L−R pairs within ST spots as evidence of physical interactions between two cell types.

### Distance of CAF−M2 colocalized spots to tumor-immune interface
The distance of a single ST spot to the tumor border is the distance of this spot to its nearest tumor border spot (Fig. 6a). Furthermore, the distance of a set of CAF−M2 colocalized spots to the tumor border was calculated by averaging the distances of all CAF−M2 spots to tumor border (green dashed line in Fig. 6b). To calculate the null distribution, we randomly selected the same number of spots from the CAF or M2-dominated spots and then calculated their distances to the tumor border.

### Pathway enrichment analysis on close and distant cancer cells to CAF−M2 interactions
Differential gene expression analysis between close and distant malignant spots to CAF−M2 interactions was performed using R package limma[50]. limma was run with or without adjusting covariate effects, i.e., malignant, CAF, M2, and CAF + M2 cell fractions. Further, the ranked gene list based on the $t$ value was used to carry out gene set enrichment analysis (GSEA) by using the R package fgsea[54]. The hallmark pathway gene sets were collected from MSigDB database[55].

### Reporting summary
Further information on research design is available in the Nature Portfolio Reporting Summary linked to this article.

## Data availability

The single-cell RNA-seq data were acquired from the following accession numbers: Non-small cell lung cancer (E-MTAB-6149 and GSE127465), Head and neck squamous cell carcinoma (GSE103322), Breast cancer (GSE114725), Melanoma (GSE115978 and GSE123139), Hepatocellular carcinoma (GSE140228), and Colorectal cancer (GSE146771). The spatial transcriptomics data were acquired from the following hyperlinks: Breast cancer ([https://www.10xgenomics.com/resources/datasets/human-breast-cancer-block-a-section-1-1-standard-1-0-0] and [https://www.10xgenomics.com/resources/datasets/human-breast-cancer-ductal-carcinoma-in-situ-invasive-carcinoma-ffpe-1-standard-1-3-0]), Glioblastoma [https://www.10xgenomics.com/resources/datasets/human-glioblastoma-whole-transcriptome-analysis-1-standard-1-2-0], Ovarian cancer [https://www.10xgenomics.com/resources/datasets/human-ovarian-cancer-whole-transcriptome-analysis-stains-dapi-anti-pan-ck-anti-cd-45-1-standard-1-2-0], Prostate cancer [https://www.10xgenomics.com/resources/datasets/human-prostate-cancer-adenocarcinoma-with-invasive-carcinoma-ffpe-1-standard-1-3-0], Squamous cell carcinoma (GSE144240), Colon cancer (SCP1278), and Pancreatic ductal adenocarcinoma (GSE111672). Details of single-cell RNA-seq and spatial transcriptomics datasets are described in Supplementary Table 2 and Supplementary Table 3, respectively. The Cancer Genome Atlas (TCGA) data are available for download at https://gdc.cancer.gov/. Hallmark gene sets (v.7.2) are available at https://www.gsea-msigdb.org. Source data are provided with this paper.

## Code availability

The source code of R package SpaCET and a demo workflow to reproduce our main results are available at GitHub (https://github.com/data2intelligence/SpaCET) and Zenodo (https://doi.org/10.5281/zenodo.7466025)[56].

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

## Acknowledgements

We would like to acknowledge our colleagues Kun Wang, Eytan Ruppin, and Prof. Jun Liu for their helpful feedback. This work is supported by the intramural research budget and FLEX award provided by the National Cancer Institute (NCI), National Institutes of Health (NIH). This study utilized the high-performance computational capabilities of the Biowulf Linux cluster at the NIH.

## Author contributions

B.R. and P.J. conceived the project and designed all studies. B.R. developed the SpaCET framework and performed computational analysis. K.A., J.H., and Y.Z. annotated the H&E image. B.R. and P.J. wrote the manuscript. All authors read and approved the final manuscript.

## Funding

## Competing interests

The authors declare no competing interests.
