## [Peer review file · Nature Communications]

REVIEWER COMMENTS

Reviewer #1 (Remarks to the Author): Expert in bioinformatics, subclone and lineage inference in cancer, and single-cell genomics

This paper proposes SPACE, a new analysis tool for spatial transcriptomics (ST) data. SPACE is specifically geared towards tumor ST data, and tackles the challenges of tumor cell detection and cell type deconvolution. There are already quite a few deconvolution methods for ST data (to estimate the proportions of each cell type at each spatial spot). However, these methods don't do well when there are unknown cell types, which is common in tumor data because tumor cells don't have a good "reference" transcriptome. Thus, I feel that this paper is addressing a real analysis gap, as existing deconvolution methods don't work well for tumor ST data.

However, the SPACE algorithm feels very ad hoc, lacking in novelty (it is basically a pipeline that strings together existing software) and a bit short in validation on real data.

The detection of malignant cell fraction is based on an existing method, inferCNV, for detecting large-scale copy number variations in each of the spots. The detection of CNVs from single cell RNA-sequencing data is a difficult problem, with few good tools, and inferCNV, although commonly used, has poor performance. SPACE computes a "malignancy score" for each spot based on the CNVs inferred by infer-CNV: If a spot has high sum of absolute CNV values across genes, then it is likely to be malignant.

Then, SPACE uses a constrained nonnegative linear regression to estimate the fractions of the immune cell types. This approach is pretty typical of existing methods and there is not much novelty.

Specifically, my concerns are:

1. Inferring CNVs from RNAseq data is known to be fickle and unreliable. Although inferCNV is commonly used, that is because there are no good alternatives, and inferCNV often has poor performance. Broad changes in gene expression can be due to differences in cell lineage.
2. Often, tumor data have accompanying bulk DNA sequencing data. Shouldn't this bulk DNA seq data be useful for CNV inference?
3. The hierarchical deconvolution method is also very specific to the data set used for illustration. It assumes that the reference marker genes have approx. 0 expression in the unknown cell types, which is very stringent. It is not clear that this method generalizes beyond the examples used.

4. More validation is needed. The method is only tested on simulations and one (!) public breast cancer sample. On the public breast cancer sample, they compared the results of SPACE to manual annotations. More testing on more samples is needed to show that the method works.

5. Also, there is not enough evidence that SPACE is sensitive in detecting subclones, as the algorithm is vague, and analysis is done on only one example data set (the same breast cancer sample). The analysis seems to be cherry-picking and explorative.

Reviewer #3 (Remarks to the Author): Expert in tumour microenvironment, cancer genomics, and spatial and single-cell genomics

This study by Ru et al. developed a computational framework, named SpaCE (Spatial Cellular Estimator), to decompose cell identities and estimate cell-cell interactions in tumor Spatial transcriptomics (ST) data. The authors compared the performance of SpaCE with other methods on ST decomposition including CIBERSORT, EPIC, and RCTD, by performing simulation analysis and double-blind pathology annotation and showed that SpaCE outperforms these approaches. In addition, they showed that SpaCE can be used to study cell-cell interactions within the same ST spot. Nowadays, it is being increasingly recognized that spatially resolved molecular measurements are important to understand cancer progression and patient response to therapy. The ST technologies are being increasingly used by the scientific community and there is an unmet need to develop novel computational tools to better characterize the spatial cellular landscape of tumor tissues. Currently, as most of the ST kits are not yet at single-cell resolution, a critical step for ST data analysis is to accurately decompose cell identities in ST spots. Therefore, this is an important study and SpaCE appears to be a useful tool with potential for wide application. There are a few ways that this manuscript could be enhanced.

1. An apparent limitation is the inability to identify malignant cells that are CNV-low or chromosomally stable, as SpaCE relies on copy number variations to predict malignant cells. However, chromosomally stable cancers are common. For example, around half of gastric adenocarcinoma (including the MSI subtype, the EBV+ subtype, and the GS subtype, see PMID: 25079317) characterized by TCGA were CNV-low or chromosomally stable. Many KRAS-mutant cancers including lung adenocarcinoma were also reported to be CNV-low or chromosomally stable. Some precancerous lesions and early-stage tumors could also be chromosomal stable. Also, it is unclear whether the CNV-low or chromosomally stable tumor cells can co-exist with chromosomally unstable tumor cells in the same ST spot or in a different region in the same section. It is therefore a major limitation of this tool.

2. As SpaCE infers malignant cell fractions based on large-scale CNVs inferred from gene expression data, its performance to predict tumor cells can be limited by the sequencing coverage, i.e. the number

of genes detected in each cell. In the real world, the FFPE samples and biopsies can yield low sequencing coverage (the median genes detected per cell could be as low as 500) and this can be very common. However, it was not examined in this study, how the difference in sequence coverage impacts the performance of SpaCE in predicting tumor cells.

3. It is also not tested how the selection of control cell types used for inferCNV will influence SpaCE's performance in predicting tumor cells. For example, macrophages usually have high expression of HLA genes on chromosome 6, which may lead to false calling of chromosome 6 deletion in tested cells. It is unknown how this is handled when there is a need to use such cells as references for CNV calling.

4. The high rate of unidentifiable cells appears to be a bit problematic. In Figure 3b, the unidentifiable cell fractions were high for stromal cells, macrophages and Lymphoid cells. Can the performance be improved in terms of better identify those "unidentifiable cells"?

5. When analyzing the spatial structure of malignant cell clones in breast cancer in Figure 4a, it is unclear how the 3 clones A, B, C were identified. It said in their manuscript that "...Clustering of inferred copy number variances (CNVs) yielded three malignant cell clones (Fig. 4a)...". However, it is not clear to me whether this is correct. Based on their dendrogram in Fig. 4a, there were two major branches (A+B as one major branch and C as another major branch), and both were further split into two subbranches, suggesting 4 subpopulations. In panel d, it is clear that ST spots of the clone C were more spreading (regionally heterogeneous) as the clone C cells localized at 3 different areas, whereas clone A and B cells were closer, and their ST spots were more localized. Consistently, the CNV profiles in their panel b and pathway analysis in panel e both indicated that the profiles of clone A and clone B were more similar.

6. In Figure 5b, I agree that the colocalization of CAFs and M2 macrophages may not be due to their profile similarity as the similarity between CAF and M2 macrophage reference profiles was low based on their correlation analysis. It is known that CAF and M2 macrophages belong to two very distinct lineages that can be easily distinguished using single-cell expression data. However, the reference profiles of CD4 naïve T cells, T helper cells, and Tregs are more similar based on their correlation analysis in Figure 5b. This may impact the level 2 deconvolution, i.e. distinguish of sub-lineages.

7. "... GSEA reported several "leading-edge" genes that are members of the EMT pathway (e.g., COL1A1, LRRC15, and LUM, Supplementary Table 2), indicating that malignant cells close to CAF-M2 interaction regions have high expression of these genes (Fig. 6c) ..". The authors have to be careful to make such a statement, as although COL1A1 is one of the genes included in the EMT pathway, it is a canonical marker gene of fibroblasts. The increased COL1A1 expression could be simply due to the presence of fibroblasts.

Reviewer #4 (Remarks to the Author): Expert in spatially resolved transcriptomics, bioinformatics, and cancer genomics

Ru and colleagues present their framework, SpaCE, to deconvolve mixed signals in plate/array-based spatially resolved transcriptomics assays to infer cell identities and interactions in cancer samples. The authors adopt a novel approach of iterative deconvolution: (i) firstly by identifying the malignant cell fraction of spots by probing for evidence of genome instability via inferCNV; (ii) supervised deconvolution of normal cell fraction with an unknown component; (iii) interrogation of spatially restricted receptor-ligand interactions. They demonstrate the method on a simulated benchmark, and some breast cancer ST datasets.

Compared to other spatial transcriptomics analytical frameworks (e.g. SquidPy and Giotto), the offerings of SpaCE are limited and have not been demonstrated on multiple spatially resolved transcriptomics assays (e.g. slide-seq). The major novelty here is the use of CNV analysis to deconvolve the tumor cell fraction before deconvolving the normal cell fraction. While the application of inferCNV to Visium data was already described by Kueckelhaus et al 2020 (<https://doi.org/10.1101/2020.10.20.346544>), this is the first of using the CNV profiles as part of a deconvolution strategy.

While the framework presents a novel approach, I believe that the work is still in a preliminary stage. I hope that the following comments will clarify the issues that I see with the current version of the manuscript.

[1] Framework Model

[1.1] The cancer cell proportion does not seem to be deconvolved in the analysis presented. Many spatial omics studies have revealed that ITH is apparent in various omics modalities. The authors point out that CNV variability can be delineated (figure 1B) in the malignant proportion, but this comes across as a side remark rather than an implicit part of the model (i.e. identify malignant clone 1, clone 2, clone 3), or their downstream analysis (e.g. co-localization or receptor-ligand analysis)

[1.2] The cell typing of the normal fraction has some limitations...

[1.2.1] Compared to the LM22 immune cell signature matrix (Cibersort paper, Newmann et al 2015) and the immune cell types in the SPOTlight paper (figure 4d, Elosua-Bayes et al 2021), the authors deconvolve fewer immune cell types (11 immune cell types, compared to 22 in LM22 and 25 in

SPOTlight). Can the authors comment on why they chose fewer immune cell types and whether they tried to achieve the same cell type resolution as LM22/SPOTlight for the immune cell types?

[1.2.2] Compared to the SPOTlight paper (figures 3a and 4a, Elosua-Bayes et al 2021), the authors deconvolve much fewer normal non-immune cell types. The cancer example presented in the SPOTlight paper (4a) 9 different normal non-immune cell types are presented, compared to 2 cell types (plus the “unidentifiable”) in this study. While I appreciate that the normal non-immune cell types will be highly variable for different types of tumors (reflecting the various organ and tissues from which tumors can be resected), if these are not deconvolved accurately then downstream analysis of co-localization and receptor-ligand interactions are lacking. The authors should rationalize their design choices, and then demonstrate equivalent/better performance to existing methods (see point 1.3).

[1.2.3] Linked to comment 1.2.2, I find the inclusion of a single CAF signature to be surprisingly few, given the large variety of tissues/organs from which tumors can be found. The authors should demonstrate that indeed a single CAF signature is sufficient for multiple tissue/organ types. The authors should rationalize their design choices, and then demonstrate equivalent/better performance to existing methods (see point 1.3).

[1.2.4] How does the “unknown” component model spatially variable cell types not in the model? In the case of multiple unknown cell types that are spatially variable, will the “unknown” component be an average expression of these cell types? If so, then this would potentially lead to a worse fit for the model (i.e. the fit would be better if there were more unknown components), and insights into co-localization patterns would be lost. Perhaps a deconvolution of the “unknown” fraction using can solve this. Alternatively, the authors could investigate having multiple “unknown” signatures if that is possible.

[1.3] Linked to point 1.1 and 1.2, the authors should demonstrate the usefulness of the tool for different tumors types in different settings by applying it to PDAC (PDAC A-STQ from Elosua-Bayes et al, <https://doi.org/10.1093/nar/gkab043>) and glioblastoma (Sample UKF269_T from Extended figure 1 of Ravi et al <https://doi.org/10.1101/2021.02.16.431475>), and compare results to the original studies, focussing of the tumor cell fraction heterogeneity, and the plethora of cells in the normal fraction.

[2] Simulation

[2.1] The benchmark setup presented in figure 2 does not seem to account for differences in the read counts of Visium vs scRNAseq experiments. The benchmark should be redone to also consider differences in read counts (e.g. 50%, 100%, and 200% of the read count average of several Visium experiments)

[2.2] The authors fail to compare their tool to many of the other spatial gene expression deconvolution tools, e.g. SPOTlight, Stereoscope (<https://doi.org/10.1038/s42003-020-01247-y>), MuSiC (<https://doi.org/10.1038/s41467-018-08023-x>), SCDC (<https://doi.org/10.1093/bib/bbz166>), cell2loc (<https://doi.org/10.1101/2020.11.15.378125>). Given the novelty of the SpaCE framework, they should at least demonstrate equivalent performance to the state of the art. Right now the comparison is limited

to RTCD. This should be at least expanded to 3 tools (acknowledging the other tools, and rationalizing their choice of which 3 they benchmark against)

[2.3] I would also be interested in a comparison to Tangram (Bianchalani et al 2020, <https://doi.org/10.1101/2020.08.29.272831>). Compared to the tools described in point 2.2, Tangram is unique in making use of the histological data to deconvolve gene expression signatures in spots. This would make an interesting comparison if SpaCE is more performant than some other tools in 2.2.

[2.4] The authors fail to compare their tool to a normal Visium workflow that does not deconvolve spots. While this should inherently be worse, the authors do not investigate this.

[3] Receptor ligand interaction

[3.1] Despite showing that there are 3 tumor subclones based on CNV clustering, the authors still only show interaction for “malignant”. The authors should redo their downstream analysis using the 3 distinct tumor subclones.

[3.2] The co-localization correlation-based approach seems to be similar to the approach described in the Stereoscope paper. The authors should compare and acknowledge this, or make differences in the approach more clear.

[3.3] The approach to interrogate potential receptor-ligand interaction (shuffling/permutation-based analysis) seems similar to cellPhoneDB and other tools which perform permutation to identify significant receptor-ligand interactions. The authors should compare and acknowledge this, or make differences in the approach more clear.

[3.4] The authors should also demonstrate the added value of the spatial information by comparing their receptor-ligand interaction analysis to that of matched scRNAseq data. This should validate the interactions that are found by SpaCE, but also highlight potential false positive in the scRNAseq data (i.e. where an interaction was predicted, but the cells do not co-localize) and false positive in the SpaCE analysis (i.e. where SpaCE predicted interactions, but such interactions cannot be recapitulated in the scRNAseq analysis)

Minor points

[4] The authors should present performance metrics, e.g. RAM and CPU time requirements.

[5] The title should make it clear that SpaCE only works for cancer.

[6] “Spatial Transcriptomics” (e.g. line 41) is an ambiguous term. It describes a specific assay, but also the general body of methods to profile spatially resolved gene expression. Please change this to another term, e.g. “Spatially resolved transcriptomics”, and only use “Spatial Transcriptomics” when referring to the specific assay.

[7] The authors should consider using the term CNA instead of CNV as this is becoming more common to cancer genomics.

[8] The formula in figure 1c could be removed without losing anything major

[9] Correlation values should be described as very high, high, moderate, and low

[10] Some of the p-value calculations are inappropriate for such large data. The authors should instead make use of effect size calculation (e.g. Cohens D).

[11] Line 38 – “genomics” should be “transcriptomics”

[12] Line 53-54 could be supported with a reference.

[13] Line 56-57 could be supported with a reference.

[14] Line 57-59. This is a strong statement, and untrue given the approach used by Tangram. Also, Stereoscope and SPOTlight “should” not be affected by cellular density if I understand their approaches correctly. The authors should investigate if their statement is indeed true or remove it.

[15] Line 163. Did the authors indeed evaluate accuracy or correlation?

[16] Line 209. The statement “effectiveness of our algorithmic designs” should be “effectiveness of our algorithmic design in a simulated setting”.

[17] After a thorough comparison to existing tools, the authors should revise which parts of their workflow are novel. To my understanding, this is limited to the use of inferCNV to identify the malignant cell proportions.

Reviewer #1

General comment a, This paper proposes SPACE, a new analysis tool for spatial transcriptomics (ST) data. SPACE is specifically geared towards tumor ST data, and tackles the challenges of tumor cell detection and cell type deconvolution. There are already quite a few deconvolution methods for ST data (to estimate the proportions of each cell type at each spatial spot). However, these methods don't do well when there are unknown cell types, which is common in tumor data because tumor cells don't have a good "reference" transcriptome. Thus, I feel that this paper is addressing a real analysis gap, as existing deconvolution methods don't work well for tumor ST data.

Response: We thank the reviewer for the appreciation of this study and all constructive suggestions.

General comment b, However, the SPACE algorithm feels very ad hoc, lacking in novelty (it is basically a pipeline that strings together existing software) and a bit short in validation on real data. The detection of malignant cell fraction is based on an existing method, inferCNV, for detecting large-scale copy number variations in each of the spots. The detection of CNVs from single cell RNA-sequencing data is a difficult problem, with few good tools, and inferCNV, although commonly used, has poor performance. SPACE computes a "malignancy score" for each spot based on the CNVs inferred by infer-CNV: If a spot has high sum of absolute CNV values across genes, then it is likely to be malignant.

Response: We agree with the reviewer that in our initial SpaCE, utilizing the existing inferCNV software might not be the optimal solution for predicting malignant cell fraction. Meanwhile, reviewers' fantastic suggestions provided better solutions to overcome the limitation of inferCNV. Thus, we have redesigned the SpaCE framework and validated the performance of our new algorithm on eight real tumor ST data across seven cancer types with double-blind annotation on matched H&E tissue images from our pathologists.

- We replaced the malignant cell inference procedure from the previous inferCNV-based approach with the correlation analysis between ST spot profiles and **cancer-type specific CNA profiles** from TCGA, suggested by the current reviewer. Such update resolved two limitations of the inferCNV approach: 1, potential chromosomal segmental transcriptomic changes due to cell lineages but not cancer CNA; 2, the extensive computation time and memory cost.
- In tumor ST profiles with low CNA, **cancer-type specific expression profiles** would substitute cancer-type specific CNA profiles. This solution fixes the defect of the inferCNV-based strategy for analyzing the CNA-low chromosomally stable tumors raised by reviewer #3.

The details of these changes are available in response to point 1 below. Inspired by reviewers' suggestions, we believe these revisions mentioned above have brought significant novelties and improvements to the SpaCE framework.

General comment c, Then, SPACE uses a constrained nonnegative linear regression to estimate the fractions of the immune cell types. This approach is pretty typical of existing methods and there is not much novelty.

Response: We admit that several existing methods also utilized nonnegative linear regression for decomposing cell types. However, we improved this regression with two strategies not utilized by previous methods to resolve challenges from the tumor ST data.

(1) Hierarchical regression for major and sub lineages to confine the collinearity of closely related cell types. SpaCE is the only framework providing a built-in comprehensive hierarchical cell-type reference in the tumor microenvironment. Although Cell2location and MuSiC have provided a similar hierarchical regression function by asking users to define cell hierarchies manually, these methods do not provide a built-in reference. Users need to find paired single-cell datasets for an ST analysis. The simulation analysis by synthetic ST data from scRNA-seq demonstrated that SpaCE with a hierarchical regression outperformed SpaCE without one on decomposing both major and sublineages (Supplementary Fig. 2e). Furthermore, SpaCE achieved better cell decomposition performances than other methods based on simulation studies (Fig. 2e).

Supplementary Fig. 2e, Deconvolution results of SpaCE with (red bar) and without (gray bar) hierarchical lineage. A dot represents a simulated ST dataset synthesized from a single scRNA-seq dataset (n=10). Each simulated ST dataset was decomposed by using a leave-one-out signature, which is the reference derived from all scRNA-seq datasets except the one left out to synthesize the simulated ST data. The y-axis presents the median Pearson correlation r between predicted and known cell fractions across cell types. The difference of groups was evaluated by the two-sided Wilcoxon rank sum test. Bar height denotes average value across simulated ST datasets; error bars denote standard errors.

Fig. 2e, Performance comparison between SpaCE and previous methods. A dot represents a simulated ST dataset synthesized from a single scRNA-seq dataset (n=10). The y value of an ST dataset presents the median Pearson correlation r between predicted and known cell fractions across cell types. All tools used the leave-one-out signature. The difference between SpaCE and other tools was evaluated by the Wilcoxon rank-sum test. Bar height denotes average value across simulated ST datasets; error bars denote standard errors.

(2) Unidentifiable components in regression to adjust unknown cell types and cellular density variations across tumor regions. This strategy is essential to correct cell fractions when the local cellular density is low, as cell counts around each ST spot will be different. For example, unidentifiable component fractions in breast cancer tissue are significantly higher among regions with low cellular densities, thus adjusting the total cell fractions (Supplementary Fig. 4c-d). Spots in low-density regions have significantly higher drop-out rates, reflected as the low unique molecular identifier (UMI) counts, than spots in high-density ones. Unidentifiable component fractions negatively correlate with the UMI counts, thus reflecting the local cellular densities.

Supplementary Fig. 4. c, H&E image with pathology annotations of high and low cellular density regions. **d,** The association between unidentifiable components and UMI counts across ST spots. All spots were grouped in the high and low cellular density regions. The values of two groups were compared by using Cohen's d effect size and two-sided Wilcoxon rank-sum test.

We evaluated this strategy on eight tumor ST datasets across seven cancer types based on double-blind annotations of the matched H&E and antibodies staining images. SpaCE with an unidentifiable component performs slightly better than SpaCE without one in stroma regions, which often have low cell density (Supplementary Fig. 13b).

Supplementary Fig. 13b. *Deconvolution results of SpaCE with and without unidentifiable components.* The annotations are the same as panel a. Each dot represents a ST dataset. The sub-panels represent the prediction of distinct regions from tumor tissue. Bar height denotes the average AUC values across ST datasets; error bars denote standard errors.

SpaCE outperformed other methods compared across various annotated tumor regions based on ST data from human clinical samples (Fig. 3f). Some previous methods, such as EPIC, RCTD, Cell2location, and Stereoscope, have an unknown component in regression. However, the EPIC framework treated this part as malignant cell fractions as the default EPIC version does not have procedures for searching cancer cell populations. The rest renormalized all cell fractions to 1 after the regression, thus removing this part in the final output.

f

Fig. 3f. *Performance comparison among methods.* Each dot represents a dataset. Y-axis presents the area under the ROC curve (AUC) value of cell fraction decompositions for each method. The sub-panels represent the results in distinct tumor regions. Bar height denotes average AUC values across datasets; error bars denote standard errors.

Specifically, my concerns are:

Comment 1. *Inferring CNVs from RNAseq data is known to be fickle and unreliable. Although inferCNV is commonly used, that is because there are no good alternatives, and inferCNV often has poor performance. Broad changes in gene expression can be due to differences in cell lineage.*

Comment 2. *Often, tumor data have accompanying bulk DNA sequencing data. Shouldn't this bulk DNA seq data be useful for CNV inference?*

Response: We agree with the reviewer that the inferCNV for estimating malignant cell abundance might have poor performance due to cell lineage difference rather than cancer CNA. This reviewer's comment #2 pointed out a solution by utilizing bulk DNA genomics data. The Cancer Genome Atlas (TCGA) project has generated CNA patterns in 30 common tumor types, reflecting cancer genetic alterations but not cell lineage differences. If an ST tumor lies in a TCGA type, we can calculate the similarity between ST profiles and bulk CNA patterns to estimate malignant cell fractions. The new

design enabled accurate cell type prediction and reduced SpaCE's dependence on the inferCNV, saving 3 ~ 4 hours of running time on each ST dataset.

Additionally, reviewer #3 mentioned a limitation of the inferCNV-based strategy on chromosomally stable tumors with low CNA. Thus, we also added an alternative inference pathway by leveraging the differential expression patterns between TCGA tumors and normal controls.

Details of our new workflow are as follows (Supplementary Fig. 1a): 1) Clustering all spots from a tumor ST dataset by hierarchical clustering. 2) Determining the malignant cell clusters whose spots have significant correlations with cancer type-specific patterns, with pattern selection rules shown in the right panel. 3) Estimating malignant cell abundance across all spots. For a tumor ST dataset, the ST-specific malignant expression profile was computed as the average expression profile among spots from malignant cell clusters in step 2. Then, the expression profile of each spot within a tumor ST dataset was correlated to the ST-specific malignant profile to infer the malignant cell fraction.

Supplementary Fig. 1a, Three steps to infer malignant cell fraction without any reference, based on a dictionary of cancer type-specific gene patterns.

Signature selection procedure in step 2

- When correlating expression profiles of ST spots to the cancer type-specific signature, we set the **cancer type-specific CNA signature** as the first option because chromosomal instability is widely considered one consistent feature of human tumors.
- Alternatively, if no spots strongly correlate with the CNA signature, the **cancer type-specific expression signature** would be activated. This situation might result from chromosomally stable cancer cells with low CNA.
- For cancer types not included in our dictionary, we created a **pan-cancer expression signature** by averaging all cancer type-specific expression signatures.

Performance comparison with other methods

We evaluated our strategy using eight tumor ST datasets on seven cancer types based on double-blind annotations of the matched H&E and antibodies staining images. SpaCE estimated the cell type fractions in tumor tissue more accurately than all existing methods included.

f

Fig. 3f, Performance comparison among methods. Each dot represents a dataset. Y-axis presents the area under the ROC curve (AUC) value of cell fraction decompositions for each method. The sub-panels represent the results in distinct tumor regions. Bar height denotes average AUC values across datasets; error bars denote standard errors.

Performance evaluation for CNA-low tumors

To evaluate our procedure for CNA-low tumors, we rerun SpaCE on ST data by directly using cancer type-specific expression signature in step 2 of Supplementary Fig. 1a. We cannot find any ST data from chromosome-stable tumors as all published ST datasets presented sufficient positive correlations between tumor cell clusters and CNA signatures. Thus, we use existing ST data as a surrogate to evaluate the CNA-low procedure. The expression signatures achieved high performance, supporting the reliability of our expression-based procedure, which will start its role for CNA-low tumors.

Supplementary Fig. 13a, Deconvolution results of SpaCE by using cancer type-specific CNA and expression signatures. Each dot represents an ST dataset. The sub-panels represent the prediction of distinct regions from tumor tissue. Bar height denotes the average AUC values across ST datasets; error bars denote standard errors.

Comment 3. *The hierarchical deconvolution method is also very specific to the data set used for illustration. It assumes that the reference marker genes have approx. 0 expression in the unknown cell types, which is very stringent. It is not clear that this method generalizes beyond the examples used.*

Response: We agree with the reviewer that the reference marker genes might indeed express in the unknown cell types. To evaluate the robustness of our algorithm, we performed simulations to explore how marker gene expression in unknown cell types may influence deconvolution results. Our simulation analysis leads to the following conclusions:

1. The result will not be affected when the number of reference markers expressed in unknown cell types is low (< 30% of genes in the reference signature).
2. Otherwise, high expression of > 30% of existing reference marker genes in unknown cell types will lead to underestimation of unknown cell types. If this happens, there should be two fixes.
2a, the unknown cell type may come from close lineage to existing cell types, thus should be classified as known cell types instead of unknown.
2b, Users should do a pre-estimation of existing cell lineages for new tissue. If SpaCE's default lineages do not comprehensively cover the repertoire in the input sample, users should input additional cell signatures, which is a new feature of our revised SpaCE framework.

Here is the detail of our simulation analysis:

We have generated expression profiles of 3 reference cell types (i.e., A, B, and C) with 500 marker genes for each type and one unknown cell type. We generated multiple unknown cell type profiles, expressing different ratios (0~40%) of reference markers (Supplementary Fig. 18a). For example, 10% means that 10% of reference marker genes from cell types A, B, and C are expressed in the unknown cell type. For each setting of unknown cell type, we mixed expression profiles of four cell types (i.e., A, B, C, and unknown) with random compositions to obtain 100 different mixtures.

Supplementary Fig. 18. Influence of reference markers expressed in unknown cell types on the SpaCE performance. a, *Simulation scheme of reference and unknown cell types.*

Then, SpaCE decomposes these mixtures by using cell-type A, B, and C as reference profiles. The figure below shows the estimated cell fraction (y-axis) versus ground truth (x-axis). Each column presents the deconvolution results of 100 mixtures generated from three reference cell types and one unknown cell type. The four rows refer to A, B, C, and unknown cell types. The title of each sub-panel shows the ratio of reference marker genes expressed in the unknown cell type.

With the increasing ratios of reference marker genes expressed in the unknown cell type (column 1~5), the fraction of reference cell types tended to be overestimated (row 1~3). Conversely, the unknown cell type was underestimated (row 4). However, the estimated cell fractions of these cell types were still positively correlated to the ground truth, indicating that the relative relationship between estimations and ground truth will not be affected. The fix to this potential issue is discussed as point 2 in our summary paragraph above.

Supplementary Fig. 18b, Deconvolution results of simulated mixtures.

Even with these potential issues, we still keep the unidentifiable component in our model. First, such a feature will at least prevent inaccuracy estimation of cell fractions when an unknown lineage does exist, and its transcriptomic profile is sufficiently different from our existing lineages. Second, this unidentifiable component can also adjust the cellular density at ST spots over low tissue density regions (see the *response to general comment c* of this reviewer). Also, SpaCE with an unidentifiable component performs better than SpaCE without one in the stroma regions, which often have low cellular density.

Supplementary Fig. 13b. Deconvolution results of SpaCE with and without unidentifiable components. The annotations are the same as panel a. Each dot represents a ST dataset. The sub-panels represent the prediction of distinct regions from tumor tissue. Bar height denotes the average AUC values across ST datasets; error bars denote standard errors.

Comment 4. More validation is needed. The method is only tested on simulations and one (!) public breast cancer sample. On the public breast cancer sample, they compared the results of SPACE to manual annotations. More testing on more samples is needed to show that the method works.

Response: As suggested by the reviewer, we have performed additional validation by using eight tumor ST datasets on seven cancer types. The following table highlights the newly added datasets.

GEO/SCP ID	Cancer Type	Platform	# Spot	# Gene	Preservation	Staining	Source
N/A	Breast cancer	Visium	3,813	22,953	FF	H&E	10x Genomics
N/A	Breast cancer	Visium	2,518	17,649	FFPE	H&E	10x Genomics
N/A	Glioblastoma	Visium	3,468	25,275	FF	H&E	10x Genomics
N/A	Ovarian cancer	Visium	3,493	24,012	FF	DAPI, Anti-CD45, Anti-PanCK	10x Genomics
N/A	Prostate cancer	Visium	4,371	16,905	FFPE	H&E	10x Genomics
GSE144240	Squamous cell carcinoma	Visium	3,650	20,255	FF	H&E	PMID: 32579974
SCP1278	Colon cancer	Slide-Seq	18,288	16,270	FF	H&E	PMID: 34912115
GSE111672	Pancreatic ductal adenocarcinoma	Old ST	428	14,574	FF	H&E	PMID: 31932730

Supplementary Table 3. Tumor spatial transcriptomics datasets used in this study. ST: Spatial transcriptomics; FF: Fresh frozen; FFPE: Formalin-fixed paraffin-embedded; H&E: Hematoxylin & Eosin

The deconvolution results were evaluated based on double-blind annotations of the matched H&E or antibodies staining images. According to the H&E morphology, pathologists labeled the regions of tumor, stroma, lymphocyte, and macrophage without knowing any deconvolution results. For ovarian cancer with antibody staining image (the 3rd highlighted line in the table above), the tumor and immune regions were labeled based on anti-PanCK and anti-CD45 signal intensity. The consistency between estimated cell fractions and double-blind pathological annotations indicates the accuracy of cell-type abundance prediction.

We compared SpaCE to ten alternative deconvolution methods for spatial and bulk RNA-seq data (Fig. 3f). All methods were run with default parameters. SpaCE consistently performed better than others, evaluated on the average AUC values in the tumor, stroma, and immune (i.e., lymphocyte, macrophage, and CD45+) regions.

f

Fig. 3f, Performance comparison among methods. Each dot represents a dataset. Y-axis presents the area under the ROC curve (AUC) value of cell fraction decompositions for each method. The sub-panels represent the results in distinct tumor regions. Bar height denotes average AUC values across datasets; error bars denote standard errors.

Comment 5. Also, there is not enough evidence that SPACE is sensitive in detecting subclones, as the algorithm is vague, and analysis is done on only one example data set (the same breast cancer sample). The analysis seems to be cherry-picking and explorative.

Response: We understand the reviewer’s concern about detecting subclones as the ST data may not directly support tumor subclonal identification. Therefore, we have made the following changes:

First, we have changed the concept from “*malignant subclone*” to “*cancer cell state*” because malignant cell clones may implicate cell branches from tumor genetic evolution through CNA and somatic mutations. However, both tumor cells’ genetic background and cell-cell interactions from the surrounding environment can determine gene expression profiles of tumor cells. Thus, the ST data may not directly support tumor sub-clonal identification.

Second, detecting cancer cell states is not the primary focus of SpaCE, which aims to decompose tumor ST data. The breast cancer example in our study shows potential downstream analyses from our deconvolution results. Due to the explorative nature, we have moved this figure from the main panel to Supplementary Figure. Meanwhile, in the revised SpaCE framework, we make this cancer cell state analysis an automatic function.

Supplements: the revised result of cancer cell state inference

In the initial submission, this analysis relies on the inferCNV results. As described in response to point 1, we have removed the dependence on inferCNV. In the new strategy, based on the deconvolution results of the breast cancer dataset, we selected the ST spots with high fractions (> 0.7) of malignant cells as tumor spots. Then, SpaCE hierarchically clustered these malignant spots to infer different states. The Silhouette value, measuring the similarity among the ST spots within each cluster compared to other clusters, was used to select the optimal cluster number. According to the silhouette analysis, the breast tumor has two cancer cell states (Supplementary Fig. 4e).

To explore cancer cell state functions, we utilized 16 consensus cancer gene modules generated from a pan-cancer scRNA-Seq analysis¹. Two cancer cell states from different spatial regions have distinct activated cancer modules, with state A expressing cell-cycle genes and state B expressing interferon responsive genes (Supplementary Fig. 4f). Two cancer cell states distributed in distinct regions were shown with spatial maps (Supplementary Fig. 4g).

Supplementary Fig. 4. **e**, Clustering and silhouette analysis of malignant spots to identify cancer cell states. The point preceding the largest decrease in silhouette value was selected as the optimal cluster number. **f**, The scores of sixteen cancer gene modules for two cancer cell states. The score of a cancer gene module for a malignant cell spot is the average expression level of all genes in this module. The score of a module for a cancer cell state is the average score of all spots belonging to this cancer cell state. **g**, Spatial distribution of two cancer cell states across the tumor tissue.

Reviewer #3

This study by Ru et al. developed a computational framework, named SpaCE (Spatial Cellular Estimator), to decompose cell identities and estimate cell-cell interactions in tumor Spatial transcriptomics (ST) data. The authors compared the performance of SpaCE with other methods on ST decomposition including CIBERSORTx, EPIC, and RCTD, by performing simulation analysis and double-blind pathology annotation and showed that SpaCE outperforms these approaches. In addition, they showed that SpaCE can be used to study cell-cell interactions within the same ST spot. Nowadays, it is being increasingly recognized that spatially resolved molecular measurements are important to understand cancer progression and patient response to therapy. The ST technologies are being increasingly used by the scientific community and there is an unmet need to develop novel computational tools to better characterize the spatial cellular landscape of tumor tissues. Currently, as most of the ST kits are not yet at single-cell resolution, a critical step for ST data analysis is to accurately decompose cell identities in ST spots. Therefore, this is an important study and SpaCE appears to be a useful tool with potential for wide application. There are a few ways that this manuscript could be enhanced.

Response: We thank the reviewer for the encouragement and constructive comments.

Comment 1. An apparent limitation is the inability to identify malignant cells that are CNV-low or chromosomally stable, as SpaCE relies on copy number variations to predict malignant cells. However, chromosomally stable cancers are common. For example, around half of gastric adenocarcinoma (including the MSI subtype, the EBV+ subtype, and the GS subtype, see PMID: 25079317) characterized by TCGA were CNV-low or chromosomally stable. Many KRAS-mutant cancers including lung adenocarcinoma were also reported to be CNV-low or chromosomally stable. Some precancerous lesions and early-stage tumors could also be chromosomal stable. Also, it is unclear whether the CNV-low or chromosomally stable tumor cells can co-exist with chromosomally unstable tumor cells in the same ST spot or in a different region in the same section. It is therefore a major limitation of this tool.

Response: The reviewer raised a fundamental limitation of inferCNV-based methods in analyzing chromosomally stable cancers. Thus, we redesigned the malignant cell prediction module (Supplementary Fig. 1a) by adding a dictionary of cancer type-specific expression signatures from TCGA.

Supplementary Fig. 1a, Three steps to infer malignant cell fraction without any reference, based on a dictionary of cancer type-specific gene patterns.

In the revised SpaCE framework, the **cancer type-specific CNA signature** of a cancer type was computed by averaging bulk tumor CNA values across patients. The **cancer type-specific expression signature** of a cancer type was generated as the differential expression profile between tumor and normal samples. When correlating the expression profiles of ST spots to the cancer type-specific signature, we set the cancer type-specific CNA signature as the first option. Among CNA-low cancer cells, if no spots have strong correlations with the CNA signature, the cancer type-specific expression signature would be activated. Additionally, we created a pan-cancer expression signature by averaging all cancer type-specific expression signatures for cancer types not included in the TCGA dictionary.

We evaluated the malignant cell inference performance using eight tumor ST datasets on seven cancer types with double-blind annotations of the matched H&E and antibody staining images. All ST data presented positive correlations between tumor spot expression and CNA signatures; thus we used these ST data as surrogates to evaluate the cancer type-specific expression signatures. We performed malignant cell inference using either CNA signatures or expression signatures (Supplementary Fig. 13a) and found that both methods achieved high and comparable performance. This result demonstrated the reliability of our expression-based signature designed for CNA-low tumors.

Supplementary Fig. 13a, *Deconvolution results of SpaCE by using cancer type-specific CNA and expression signatures*. Each dot represents an ST dataset. The sub-panels represent the prediction of distinct regions from tumor tissue. Bar height denotes the average AUC values across ST datasets; error bars denote standard errors.

Comment 2. *As SpaCE infers malignant cell fractions based on large-scale CNVs inferred from gene expression data, its performance to predict tumor cells can be limited by the sequencing coverage, i.e. the number of genes detected in each cell. In the real world, the FFPE samples and biopsies can yield low sequencing coverage (the median genes detected per cell could be as low as 500) and this can be very common. However, it was not examined in this study, how the difference in sequence coverage impacts the performance of SpaCE in predicting tumor cells.*

Response: The reviewer brings up an important point about whether the preservation method of tumor samples impacts the sequencing coverage of ST data and further SpaCE's performance.

In the revised analysis, two of eight datasets are FFPE samples from breast (Supplementary Fig. 5b) and prostate (Supplementary Fig. 8b) tumors, respectively. We first checked the median Unique Molecular Identifier (UMI) and gene counts across ST spots. Both datasets have >10,000 median UMI counts and >4,000 median genes per ST spot, which are similar to the fresh-frozen samples from another breast cancer (Supplementary Fig. 4a) and a glioblastoma (Supplementary Fig. 6b).

Supplementary Fig. 5b, Histogram of UMI counts and genes per spot. The dashed line represents the median value across all spots.

Supplementary Fig. 8b, Histogram of UMI counts and genes per spot. The dashed line represents the median value across all spots.

Supplementary Fig. 4a, Histogram of UMI counts and genes per spot. The dashed line represents the median value across all spots.

Supplementary Fig. 6b, Histogram of UMI counts and genes per spot. The dashed line represents the median value across all spots.

Spatial transcriptomics techniques on fresh-frozen and FFPE samples utilize totally different techniques. As described in the 10x official website², Visium FFPE workflow uses a pair of complementary DNA probes to hybridize on the whole transcriptome to amplify signals, instead of relying on cRNA reverse transcription that will be affected by RNA qualities. Thus, Visium FFPE may not have low gene coverage issues.

Subsequently, we evaluated how the sequence coverage impacts the SpaCE performance in predicting malignant cells by downsampling the gene counts per spot on two FF and two FFPE samples. The performance is stable in both FFPE datasets upon down-sampling read counts until 500 genes per spot. Therefore, we reason that the analysis quality on FFPE samples will not be an issue if the data is appropriately generated, especially with the rapid improvement of ST technologies.

Supplementary Fig. 13c, *Deconvolution results of SpaCE on the randomly downsampled ST datasets*. The gene counts per spot were sampled from 4000 to 500 genes. Each dot represents a randomization replicate (10 dots in total). Bar height denotes the average AUC values across ten randomizations; error bars denote standard errors.

Comment 3. It is also not tested how the selection of control cell types used for inferCNV will influence SpaCE's performance in predicting tumor cells. For example, macrophages usually have high expression of HLA genes on chromosome 6, which may lead to false calling of chromosome 6 deletion in tested cells. It is unknown how this is handled when there is a need to use such cells as references for CNV calling.

Response: The reviewer has raised an essential concern about the influence of control cells on inferCNV, particularly when chromosome-level expression changes happen due to cell lineage differences. In the revised SpaCE, we removed inferCNV and thus avoided dependence on the control cell selection. Alternatively, we have built a dictionary of cancer type-specific CNA signatures for 30 tumor types from TCGA, defined as gene-level copy number difference from diploid (Supplementary Fig. 1c and Supplementary Table 1, also see response to comment 1 of the same reviewer). These CNA signatures from bulk tumors reflect cancer genetic alterations instead of cell lineage differences.

Supplementary Fig. 1c, Cancer type-specific CNA dictionary for TCGA cancer types, computed as the average CNA values (y-axis) over chromosomal locations (x-axis) across patients.

Cancer	Full Name	CNA	Expression
ACC	Adrenocortical carcinoma	√	
BLCA	Bladder Urothelial Carcinoma	√	√
BRCA	Breast invasive carcinoma	√	√
CESC	Cervical squamous cell carcinoma and endocervical adenocarcinoma	√	
CHOL	Cholangiocarcinoma	√	
COAD	Colon adenocarcinoma	√	√
ESCA	Esophageal carcinoma	√	√
GBM	Glioblastoma multiforme	√	
HNSC	Head and Neck squamous cell carcinoma	√	√
KICH	Kidney Chromophobe	√	√
KIRC	Kidney renal clear cell carcinoma	√	√
KIRP	Kidney renal papillary cell carcinoma	√	√
LGG	Brain Lower Grade Glioma	√	
LIHC	Liver hepatocellular carcinoma	√	√
LUAD	Lung adenocarcinoma	√	√
LUSC	Lung squamous cell carcinoma	√	√
MESO	Mesothelioma	√	
OV	Ovarian serous cystadenocarcinoma	√	
PAAD	Pancreatic adenocarcinoma	√	
PCPG	Pheochromocytoma and Paraganglioma	√	
PRAD	Prostate adenocarcinoma	√	√
READ	Rectum adenocarcinoma	√	√
SARC	Sarcoma	√	
SKCM	Skin Cutaneous Melanoma	√	
STAD	Stomach adenocarcinoma	√	√
TGCT	Testicular Germ Cell Tumors	√	
THCA	Thyroid carcinoma	√	√
UCEC	Uterine Corpus Endometrial Carcinoma	√	√
UCS	Uterine Carcinosarcoma	√	
UVM	Uveal Melanoma	√	
PANCAN	Pan-cancer		√

Supplementary Table 1. A gene pattern dictionary of copy number alterations (CNA) and tumor-normal expression differences generated from The Cancer Genome Atlas (TCGA). Certain tumor types do not have normal tissue controls, thus, we do not generate tumor-normal profiles for them.

We evaluated our strategy using eight tumor ST datasets on seven cancer types based on double-blind annotations of the matched H&E and antibody staining images. SpaCE estimated the cell type fractions in tumor tissue more accurately than existing methods (Supplementary Fig. 12 and Fig. 3f).

Supplementary Fig. 12. Deconvolution results of eight ST datasets on seven cancer types.

The AUC values of SpaCE and alternative methods for predicting different cell types (column) across ST datasets (row). The dashed line represents AUC = 0.5 as the random expectation. AUC: Area under the ROC Curve; ROC: Receiver Operating Characteristic; FF: Fresh frozen; FFPE: Formalin-fixed paraffin-embedded; ST: spatial transcriptomics.

Fig. 3f, Performance comparison among methods. Each dot represents a dataset. Y-axis presents the area under the ROC curve (AUC) value of cell fraction decompositions for each method. The sub-panels represent the results in distinct tumor regions. Bar height denotes average AUC values across datasets; error bars denote standard errors.

Comment 4. *The high rate of unidentifiable cells appears to be a bit problematic. In Figure 3b, the unidentifiable cell fractions were high for stromal cells, macrophages and Lymphoid cells. Can the performance be improved in terms of better identify those “unidentifiable cells”?*

Response: We apologize for the confusion on the unidentifiable component and have clarified it in the revision. The unidentifiable component reflects two signals: (1) the existence of unknown cell types not included in our reference. (2) the noise due to low cellular density in a tumor tissue region. The SpaCE regression does not require the sum of cell fractions (coefficients) as 1. Thus, if a significant amount of noise exists on an ST spot due to a low tissue density, the regression process will reduce cell fractions by increasing the unidentifiable component coefficient.

For example, in a breast tumor ST profile, low-density regions in the stroma have higher unknown components. Spots in low-density regions have significantly higher drop-out rates, reflected as the low unique molecular identifier (UMI) counts, than spots in high-density ones. Unidentifiable component fractions negatively correlate with the UMI counts, thus reflecting the local cellular densities.

Supplementary Fig. 4. c, H&E image with pathology annotations of high and low cellular density regions. **d**, The association between unidentifiable components and UMI counts across ST spots. All spots were grouped in the high and low cellular density regions. The values of two groups were compared by using Cohen's d effect size and two-sided Wilcoxon rank-sum test.

Comment 5. *When analyzing the spatial structure of malignant cell clones in breast cancer in Figure 4a, it is unclear how the 3 clones A, B, C were identified. It said in their manuscript that "...Clustering of inferred copy number variances (CNVs) yielded three malignant cell clones (Fig. 4a)...". However, it is not clear to me whether this is correct. Based on their dendrogram in Fig. 4a, there were two major branches (A+B as one major branch and C as another major branch), and both were further split into two subbranches, suggesting 4 subpopulations. In panel d, it is clear that ST spots of the clone C were more spreading (regionally heterogeneous) as the clone C cells localized at 3 different areas, whereas clone A and B cells were closer, and their ST spots were more localized. Consistently, the CNV profiles in their panel b and pathway analysis in panel e both indicated that the profiles of clone A and clone B were more similar.*

Response: We understand the reviewer's concern about detecting subclones in our initial manuscript. We have changed the concept "malignant subclone" to "cancer cell state". This is because malignant cell clones may implicate cell branches from tumor genetic evolution due to somatic mutations or CNA, while transcriptomics profiles are determined by both tumor cells' genetic background and impacts from the surrounding environment. Thus, spatial transcriptomics may not directly enable the inference of malignant cell clones.

Also, we revised the identification of cancer cell states (previously named cell clones) because our new malignant cell prediction in SpaCE no longer uses the inferCNV (response to comment 3 of this reviewer). Based on the deconvolution results of the breast cancer dataset, we selected the ST spots with high abundant malignant cells (fraction > 0.7). Then, SpaCE hierarchically clustered these malignant spots to infer different states.

The Silhouette value, measuring the similarity among ST spots within each cluster compared to other clusters, was used to select the optimal cluster number. The point preceding the largest decrease in silhouette value was selected as the optimal cluster number. This cutoff aims to capture distinct cancer cell states (by selecting a high cluster count) but avoid splitting any tight clusters (by avoiding large Silhouette drop). For example, the breast cancer ST data has two cancer cell states with this new procedure (Supplementary Fig. 4e).

To explore cancer cell state functions, we utilized 16 consensus cancer gene modules generated from a pan-cancer scRNA-Seq analysis¹. Two cancer cell states from different spatial regions have distinct activated cancer modules, with state A expressing cell-cycle genes and state B expressing interferon responsive genes (Supplementary Fig. 4f). The cancer cell states were shown in spatial maps (Supplementary Fig. 4g).

Supplementary Fig. 4. e, Clustering and silhouette analysis of malignant spots to identify cancer cell states. The point preceding the largest decrease in silhouette value was selected as the optimal cluster number. **f**, The scores of sixteen cancer gene modules for two cancer cell states. The score of a cancer gene module for a malignant cell spot is the average expression level of all genes in this module. The score of a module for a cancer cell state is the average score of all spots belonging to this cancer cell state. **g**, Spatial distribution of two cancer cell states across the tumor tissue.

Comment 6. In Figure 5b, I agree that the colocalization of CAFs and M2 macrophages may not be due to their profile similarity as the similarity between CAF and M2 macrophage reference profiles was low based on their correlation analysis. It is known that CAF and M2 macrophages belong to two very distinct lineages that can be easily distinguished using single-cell expression data. However, the reference profiles of CD4 naïve T cells, T helper cells, and Tregs are more similar based on their correlation analysis in Figure 5b. this may impact the level 2 deconvolution, i.e. distinguish of sub-lineages.

Response: We agree with the reviewer that deconvolving close sub-lineages is challenging because of their similar reference profiles. As expected in Fig. 2c, the prediction accuracy of sub-lineages (level 2) is lower than major lineages (level 1) in simulation analysis. This performance reduction on sub-lineages is a challenge that all evaluated deconvolution tools would face. However, compared to other approaches, our tool SpaCE consistently achieved a better performance in both major lineage and sublineage (Fig. 2e).

Fig. 2c, Performance in intra-dataset validation for each scRNA-seq dataset (row) and cell type (column). The color in the heatmap presents Pearson correlations (r) between predicted versus known cell fractions. The gray color in the heatmap indicates missing cell types in the scRNA-seq dataset. Box plots on the right present r values of all cell type predictions in the same dataset. Box plots on the top present r values of the same cell type across all datasets. We shuffled spot identities of cell type fraction vectors within each synthetic ST data and computed r values as random controls. For the boxplot, the thick line represents the median value. The bottom and top of the boxes are the 25th and 75th percentiles (interquartile range). The whiskers encompass 1.5 times the interquartile range.

Fig. 2e, Performance comparison between SpaCE and previous methods. A dot represents a simulated ST dataset synthesized from a single scRNA-seq dataset ($n=10$). The y value of an ST dataset presents the median Pearson correlation r between predicted and known cell fractions across cell types. All tools used the leave-one-out signature. The difference between SpaCE and other tools was evaluated by the Wilcoxon rank-sum test. Bar height denotes average value across simulated ST datasets; error bars denote standard errors.

Comment 7. "... GSEA reported several "leading-edge" genes that are members of the EMT pathway (e.g., COL1A1, LRRC15, and LUM, Supplementary Table 2), indicating that malignant cells close to CAF-M2 interaction regions have high expression of these genes (Fig. 6c) ..". the authors have to be careful to make such a statement, as although COL1A1 is one of the genes included in the EMT pathway, it is a canonical marker gene of fibroblasts. the increased COL1A1 expression could be simply due to the presence of fibroblasts.

Response: When testing whether COL1A1 expression comes from cancer cells or fibroblasts, we have adjusted for tumor, CAF, and M2 fractions during the differential expression analysis. COL1A1 still ranks as the top gene in the leading-edge lists in all analyses. Corroborating with our result, previous studies demonstrated that COL1A1 expression in cancer cells promotes cell migration in vitro in breast cancer³ and colon cancer⁴, and serves as a cancer-cell marker of epithelial-mesenchymal transition¹.

Despite these results, we still admit that we do not have rigorous experimental evidence to demonstrate the source of COL1A1 and thus have revised the statement as "... GSEA reported several "leading-edge" genes that are members of the EMT pathway (e.g., COL1A1, LRRC15, and LUM), indicating that malignant cells close to CAF-M2 interaction regions have high expression of these genes (Fig. 6c). However, additional experimental evidence is still needed to validate the gene functions in promoting cancer aggression".

Reviewer #4

Ru and colleagues present their framework, SpaCE, to deconvolve mixed signals in plate/array-based spatially resolved transcriptomics assays to infer cell identities and interactions in cancer samples. The authors adopt a novel approach of iterative deconvolution: (i) firstly by identifying the malignant cell fraction of spots by probing for evidence of genome instability via inferCNV; (ii) supervised deconvolution of normal cell fraction with an unknown component; (iii) interrogation of spatially restricted receptor-ligand interactions. They demonstrate the method on a simulated benchmark, and some breast cancer ST datasets.

General comment a, *Compared to other spatial transcriptomics analytical frameworks (e.g. SquidPy and Giotto), the offerings of SpaCE are limited*

Response: We agree with the reviewer that both SquidPy and Giotto provide many function modules, such as visualizing spatial data, computing spatial patterns, analyzing neighborhoods, and decomposing cell identities through Tangram and SpatialDWLS, respectively. Here, we clarify that the primary function of SpaCE is to deconvolve ST data. Its counterparts would be the deconvolution tools, such as Tangram and SpatialDWLS integrated into SquidPy and Giotto. Thus, in the comparison section, we have compared SpaCE to ten alternative tools, including Tangram, SpatialDWLS, RCTD, SPOTlight, *et al.*, and demonstrated the superior performance of SpaCE (response to points 1.3 & 2.2 of this reviewer).

Nevertheless, we fully share the reviewer's opinion about offering more analytical modules in our tool SpaCE. In future updates, we will certainly develop more customized functionalities for data analysis and visualization.

General comment b, *and have not been demonstrated on multiple spatially resolved transcriptomics assays (e.g. slide-seq).*

Response: In the revised manuscript, we include seven datasets from the plate or array-based ST/Visium platform and one from the bead-based Slide-Seq platform (response to major point 1.3 and minor point 4).

General comment c, *The major novelty here is the use of CNV analysis to deconvolve the tumor cell fraction before deconvolving the normal cell fraction. While the application of inferCNV to Visium data was already described by Kueckelhaus et al 2020 (<https://doi.org/10.1101/2020.10.20.346544>), this is the first of using the CNV profiles as part of a deconvolution strategy. While the framework presents a novel approach, I believe that the work is still in a preliminary stage.*

Response: We thank the reviewer for recognizing the novelty in inferCNV-based prediction of tumor cell fraction in ST data and agree that the initial workflow was not fully developed. The other two reviewers have pointed out limitations of inferCNV in identifying malignant cell populations, including 1) the influence of control cells on inferCNV results as cell lineage difference may induce chromosomal transcriptomic variations, and 2) the failure of inferCNV in analyzing chromosomally stable cancers. Therefore, we redesigned the SpaCE to address these limitations and fully develop our malignant cell inference for both CNA-high and chromosomal stable tumors with low CNA.

The core idea is to create a dictionary of cancer type-specific CNA and expression signatures using bulk tumor data from the Cancer Genome Atlas (TCGA) project, and then calculate the similarity of the ST profiles with dictionary signatures to estimate the malignant cell abundance.

Supplementary Fig. 1a, Three steps to infer malignant cell fraction without any reference, based on a dictionary of cancer type-specific gene patterns. 1) Clustering all spots from a tumor ST dataset by hierarchical clustering. 2) Determining the malignant cell clusters whose spots have significant correlations with cancer type-specific patterns, with pattern selection rules shown in the right. 3) Estimating malignant cell abundance across all spots. For a tumor ST dataset, the ST-specific malignant expression profile was computed as the average expression profile among spots from malignant cell clusters in step 2. Then, the expression profile of each spot within a tumor ST dataset was correlated to the ST-specific malignant profile to infer the malignant cell fraction.

In step2, the **cancer type-specific CNA signature** of a cancer type was computed by averaging bulk tumor CNA values across patients. The **cancer type-specific expression signature** of a cancer type was generated as the differential expression profile between tumor and normal samples.

- When correlating ST spot expression profiles with the cancer type-specific signature, we set the **cancer type-specific CNA signature** as the first option because chromosomal instability is widely considered one consistent feature of human tumors.
- Alternatively, if no spots have strong correlations with the CNA signatures, the **cancer type-specific expression signature** would be activated. This situation might result from chromosomally stable cancer cells.
- We also created a **pan-cancer expression signature** by averaging all cancer type-specific expression signatures for cancer types not included in our dictionary of cancer type-specific signatures.

We demonstrated our strategy by using eight tumor spatially resolved transcriptomics datasets on seven cancer types based on double-blind annotations of the matched H&E and antibody staining images. In the performance comparison, SpaCE outperformed all previous methods collected.

Fig. 3f, Performance comparison among methods. Each dot represents a dataset. Y-axis presents the area under the ROC curve (AUC) value of cell fraction decompositions for each method. The sub-panels represent the results in distinct tumor regions. Bar height denotes average AUC values across datasets; error bars denote standard errors.

I hope that the following comments will clarify the issues that I see with the current version of the manuscript.

Response: We thank the reviewer for constructive comments that significantly helped our study.

[1] Framework Model

[1.1] The cancer cell proportion does not seem to be deconvolved in the analysis presented. Many spatial omics studies have revealed that ITH is apparent in various omics modalities. The authors point out that CNV variability can be delineated (figure 1B) in the malignant proportion, but this comes across as a side remark rather than an implicit part of the model (i.e. identify malignant clone 1, clone 2, clone 3), or their downstream analysis (e.g. co-localization or receptor-ligand analysis)

Response: In response to the reviewer’s comment, we have made the following changes:

- In the revision, we have changed the concept “malignant subclone” as “cancer cell state”. This update is because malignant cell clones implicate cell branches from tumor genetic evolution due to either CNA or somatic mutations, while transcriptomics profiles are determined by both tumor cells’ genetic background and impacts from the surrounding environment. Thus, ST data may not directly infer cancer genetic clones.
- The cancer cell state analysis is indeed an explorative side mark and not the major focus of SpaCE. The breast cancer example was used to show the potential extension of our deconvolution results. Due to the explorative nature, we have moved this figure from the main panel to the Extended Data Figure. Meanwhile, we make the cancer cell state analysis an automatic function of the revised SpaCE framework, so users can run these extensions directly on their deconvolution results.

Since we have redesigned the malignant cell prediction procedure and removed the dependency on inferCNV (response to general comment c of this reviewer), we performed the cancer cell state

analysis with a new strategy. Based on the deconvolution results of the breast cancer dataset, we selected the ST spots with high fractions (> 0.7) of malignant cells across tumor tissue. Then, SpaCE hierarchically clustered these malignant spots to infer different states. The Silhouette value, measuring the similarity among ST spots within each cluster compared to other clusters, was used to select the optimal cluster number. The point preceding the largest decrease in silhouette value was selected as the optimal cluster number. This cutoff aims to capture distinct cancer cell states (by selecting a high cluster count) but avoid splitting any tight clusters (by avoiding large Silhouette drop). According to the silhouette analysis, the breast tumor has two cancer cell states (Supplementary Fig. 4e).

To explore cancer cell state functions, we utilized 16 consensus cancer gene modules generated from a pan-cancer scRNA-Seq analysis¹. Two cancer cell states from different spatial regions have distinct activated cancer modules, with state A expressing cell-cycle genes and state B expressing interferon responsive genes (Supplementary Fig. 4f). Similar to deconvolving immune sublineages, the SpaCE extension module can also estimate the fraction of distinct cancer cell states, using reference profiles as the average expression vector across all ST spots for each state (Supplementary Fig. 4g).

Supplementary Fig. 4. e, Clustering and silhouette analysis of malignant spots to identify cancer cell states. The point preceding the largest decrease in silhouette value was selected as the optimal cluster number. **f**, The scores of sixteen cancer gene modules for two cancer cell states. The score of a cancer gene module for a malignant cell spot is the average expression level of all genes in this module. The score of a module for a cancer cell state is the average score of all spots belonging to this cancer cell state. **g**, Spatial distribution of two cancer cell states across the tumor tissue.

[1.2] The cell typing of the normal fraction has some limitations...

[1.2.1] Compared to the LM22 immune cell signature matrix (Cibersort paper, Newmann et al 2015) and the immune cell types in the SPOTlight paper (figure 4d, Elosua-Bayes et al 2021), the authors deconvolve fewer immune cell types (11 immune cell types, compared to 22 in LM22 and 25 in SPOTlight). Can the authors comment on why they chose fewer immune cell types and whether they tried to achieve the same cell type resolution as LM22/SPOTlight for the immune cell types?

Response: We thank the reviewer for this comment. In the previous manuscript, we mainly focused on the major lineage of immune cells, and thus had fewer immune cell types. Currently, we used SingleR⁵ to recognize the sublineage of major cell types (e.g., B cell and cDC), and the hierarchical references have been expanded to include 12 major immune lineages and 20 sub-lineages (Fig. 1c).

Fig. 1c, Hierarchical deconvolution of non-malignant cell fractions.

[1.2.2] Compared to the SPOTlight paper (figures 3a and 4a, Elosua-Bayes et al 2021), the authors deconvolve much fewer normal non-immune cell types. The cancer example presented in the SPOTlight paper (4a) 9 different normal non-immune cell types are presented, compared to 2 cell types (plus the “unidentifiable”) in this study. While I appreciate that the normal non-immune cell types will be highly variable for different types of tumors (reflecting the various organ and tissues from which tumors can be resected), if these are not deconvolved accurately then downstream analysis of co-localization and receptor-ligand interactions are lacking. The authors should rationalize their design choices, and then demonstrate equivalent/better performance to existing methods (see point 1.3).

Response: The reviewer raised an important concern about normal non-immune cells, which are not sufficiently included in the built-in SpaCE reference designed for general analysis across tumor types. As the reviewer mentioned, normal non-immune cell types will be highly variable for different tumor types. It is hard to build a comprehensive normal reference. Also, many tumor ST data do not have matched scRNA-seq data. Thus, the SpaCE reference included the two most abundant normal non-immune cell types, including CAF and endothelial cells.

We admit that this design might miss other normal cell types. Following the reviewer’s comment, we designed a new function, which accepts hierarchical reference profiles customized for each tissue type (Supplementary Fig. 14a). Based on the same data in the SPOTlight study, the new module successfully decomposed relevant normal cell types at their annotated locations, including acinar and ductal cells (Supplementary Fig. 14c).

Supplementary Fig. 14. SpaCE results of pancreatic ductal adenocarcinoma ST data with matched scRNA-seq dataset. a, Hierarchical tree of cell types from the matched scRNA-seq data set. c, Pathology annotations of the H&E-stained tissue image from the original publications, matched with SpaCE deconvolution results on relevant cell types.

[1.2.3] Linked to comment 1.2.2, I find the inclusion of a single CAF signature to be surprisingly few, given the large variety of tissues/organs from which tumors can be found. The authors should demonstrate that indeed a single CAF signature is sufficient for multiple tissue/organ types. The authors should rationalize their design choices, and then demonstrate equivalent/better performance to existing methods (see point 1.3).

Response: We agree that cancer-associated fibroblasts (CAF) could be varied across tumors. However, in our hierarchical tree, they are more similar to each other when compared to other lineages (Fig. 2b). Thus, CAFs from different tumor types are classified as one CAF profile.

Fig. 2b, Hierarchical clustering of cell lineage reference profiles from scRNA-seq datasets, based on marker gene set similarities.

[1.2.4] How does the “unknown” component model spatially variable cell types not in the model? In the case of multiple unknown cell types that are spatially variable, will the “unknown” component be an average expression of these cell types? If so, then this would potentially lead to a worse fit for the model (i.e. the fit would be better if there were more unknown components), and insights into co-localization patterns would be lost. Perhaps a deconvolution of the “unknown” fraction using can solve this. Alternatively, the authors could investigate having multiple “unknown” signatures if that is possible.

Response: We apologize for the lack of clarity on the definition of the unknown component. When decomposing non-malignant cell fractions, the “unidentifiable” component models 1) the fraction of cell types not included in our hierarchical reference and 2) noise caused by low tissue density. This part of the unknown component does not have a reference profile for any cell type.

Our constrained linear regression model sets the sum of all coefficients ≤ 1 . Thus, if the expression of an ST spot contains patterns from cell types not included in the reference signature or a significant portion of random noise, the SpaCE regression will reduce the sum of fractions over known cell lineages. For example, low-density regions in a breast tumor have more unknown components than regions with high cell densities. The reason is that ST spots in low-density regions have low counts of unique molecular identifiers (UMI). Unidentifiable component fractions negatively correlate with UMI counts, thus reflecting the cellular density of local regions.

Supplementary Fig. 4. c, H&E image with pathology annotations of high and low cellular density regions. **d,** The association between unidentifiable components and UMI counts across ST spots. All spots were grouped in the high and low cellular density regions. The values of two groups were compared by using Cohen's d effect size and two-sided Wilcoxon rank-sum test.

[1.3] Linked to point 1.1 and 1.2, the authors should demonstrate the usefulness of the tool for different tumors types in different settings by applying it to PDAC (PDAC A-STQ from Elosua-Bayes et al, <https://doi.org/10.1093/nar/gkab043>) and glioblastoma (Sample UKF269_T from Extended figure 1 of Ravi et al <https://doi.org/10.1101/2021.02.16.431475>), and compare results to the original studies, focussing of the tumor cell fraction heterogeneity, and the plethora of cells in the normal fraction.

Response: We agree with the reviewer that additional validation would be valuable. We have performed additional validation using seven tumor Visium/ST datasets and one slide-seq dataset across seven cancer types based on double-blind annotations of the matched H&E or antibody staining images. Since the GBM dataset mentioned by the reviewer is from a BioRxiv paper that did

not provide a public link, we include another GBM dataset from 10x Genomics. The following table highlights the newly added datasets.

GEO/SCP ID	Cancer Type	Platform	# Spot	# Gene	Preservation	Staining	Source
N/A	Breast cancer	Visium	3,813	22,953	FF	H&E	10x Genomics
N/A	Breast cancer	Visium	2,518	17,649	FFPE	H&E	10x Genomics
N/A	Glioblastoma	Visium	3,468	25,275	FF	H&E	10x Genomics
N/A	Ovarian cancer	Visium	3,493	24,012	FF	DAPI, Anti-CD45, Anti-PanCK	10x Genomics
N/A	Prostate cancer	Visium	4,371	16,905	FFPE	H&E	10x Genomics
GSE144240	Squamous cell carcinoma	Visium	3,650	20,255	FF	H&E	PMID: 32579974
SCP1278	Colon cancer	Slide-Seq	18,288	16,270	FF	H&E	PMID: 34912115
GSE111672	Pancreatic ductal adenocarcinoma	Old ST	428	14,574	FF	H&E	PMID: 31932730

Supplementary Table 3. Tumor spatial transcriptomics datasets used in this study. ST: Spatial transcriptomics; FF: Fresh frozen; FFPE: Formalin-fixed paraffin-embedded; H&E: Hematoxylin & Eosin

According to the H&E morphology, pathologists labeled the regions of the tumor, stroma, lymphocyte, and macrophage without knowing any deconvolution results. For ovarian cancer with antibody staining images, the tumor and immune regions were labeled based on anti-PanCK and anti-CD45 signal intensity, respectively. The consistency between estimated cell fractions and double-blind annotations indicates the accuracy of cell abundance prediction. We compared SpaCE to ten alternative methods run with default parameters. In general, SpaCE yielded more accurate estimates across cell types than other methods (Fig. 3f and Supplementary Fig. 12).

Fig. 3f, Performance comparison among methods. Each dot represents a dataset. Y-axis presents the area under the ROC curve (AUC) value of cell fraction decompositions for each method. The sub-panels represent the results in distinct tumor regions. Bar height denotes average AUC values across datasets; error bars denote standard errors.

Supplementary Fig. 12. Deconvolution results of eight ST datasets on seven cancer types.

The AUC values of SpACE and alternative methods for predicting different cell types (column) across ST datasets (row). The dashed line represents AUC = 0.5 as the random expectation. AUC: Area under the ROC Curve. ROC: Receiver Operating Characteristic; FF: Fresh frozen; FFPE: Formalin-fixed paraffin-embedded; ST: spatial transcriptomics.

[2] Simulation

[2.1] The benchmark setup presented in figure 2 does not seem to account for differences in the read counts of Visium vs scRNAseq experiments. The benchmark should be redone to also consider differences in read counts (e.g. 50%, 100%, and 200% of the read count average of several Visium experiments).

Response: We agree with the reviewer's concern and thus evaluated the effect of read count differences of synthetic ST versus scRNAseq on SpaCE performance. The performance is robust and only decreases significantly at < 50% of the read count from single-cell data.

To create read count differences between synthetic ST versus single-cell datasets, we randomly downsampled the read count of each single-cell reference profile to 15%, 25%, 50%, and 75% of its original read count. For the difference of 100%, 150%, or 200%, the read count of each single-cell reference profile was multiplied by 1, 1.5, or 2 for each expressed gene, and read counts for non-expressed genes (drop out) will not be over-sampled. A drop in performance only happens at < 50% difference in the read count (Supplementary Fig. 2c).

Supplementary Fig. 2c. Impact of the read count differences of ST vs scRNAseq data. Each dot represents a simulated ST dataset synthesized from a single scRNA-seq dataset ($n=10$). The y axis presents the median Pearson correlation r and RMSE between predicted and known cell fractions across all cell types. The difference of groups was evaluated by the Kruskal-Wallis test. Bar height denotes average value across simulated ST datasets; error bars denote standard errors.

[2.2] The authors fail to compare their tool to many of the other spatial gene expression deconvolution tools, e.g. SPOTlight, Stereoscope (<https://doi.org/10.1038/s42003-020-01247-y>), MuSiC (<https://doi.org/10.1038/s41467-018-08023-x>), SCDC (<https://doi.org/10.1093/bib/bbz166>), cell2loc (<https://doi.org/10.1101/2020.11.15.378125>). Given the novelty of the SpaCE framework, they should at least demonstrate equivalent performance to the state of the art. Right now the comparison is limited to RTCD. This should be at least expanded to 3 tools (acknowledging the other tools, and rationalizing their choice of which 3 they benchmark against)

Response: The concern raised by the reviewer is legitimate. In the revision, we compared SpaCE against ten alternative methods, including those mentioned by the reviewer. SpaCE consistently performed better than other approaches in both major lineage and sublineage in simulation (Fig. 2e and Supplementary Fig. 3a-b). Moreover, SpaCE outperformed other approaches on real ST data from human tumors, evaluated as the performance of predicting double-blind pathology annotations (response to comment 1.3 of this reviewer).

Fig. 2e, Performance comparison between SpaCE and previous methods. A dot represents a simulated ST dataset synthesized from a single scRNA-seq dataset ($n=10$). The y value of an ST dataset presents the median Pearson correlation r between predicted and known cell fractions across cell types. All tools used the leave-one-out signature. The difference between SpaCE and other tools was evaluated by the Wilcoxon rank-sum test. Bar height denotes average value across simulated ST datasets; error bars denote standard errors.

[2.3] I would also be interested in a comparison to Tangram (Bianchalani et al 2020, <https://doi.org/10.1101/2020.08.29.272831>). Compared to the tools described in point 2.2, Tangram is unique in making use of the histological data to deconvolve gene expression signatures in spots. This would make an interesting comparison if SpaCE is more performant than some other tools in 2.2.

Response: The reviewer raises a question of whether integrating histological data would improve the decomposition of ST data. Tangram can incorporate cell segmentations of H&E images into their deconvolution model. However, Tangram itself does not perform cell segmentation. Instead, Tangram used pre-existing machine learning tools. A significant challenge is that cell segmentation algorithms often fail to annotate cells within higher-density regions because cells in these regions would overlap and huddle together. As the author of Tangram mentioned in their paper:

“Tangram assumed that cells are segmented in preprocessing, which we performed here with dedicated external tools (for example, ilastik or nucleAIzer). However, in higher-density tissues, such as embryos or tumors, cell-segmentation methods may not perform as well.”

To test the performance, we first performed cell segmentation for a breast tumor H&E image by ilastik that Tangram used. The segmentation performance was worse in the tumor and lymphocyte regions with high cell densities. Therefore, coupling cell segmentations compromised the deconvolution performance. We only include Tangram without the histological information in our comparisons (see response to comment 1.3 of this reviewer).

[2.4] The authors fail to compare their tool to a normal Visium workflow that does not deconvolve spots. While this should inherently be worse, the authors do not investigate this.

Response: We did not compare our tool SpaCE to normal Visium workflows because the Loupe Browser from 10x does not have the function to evaluate the proportion of a cell type across ST spots. Instead, users may use a marker gene expression to represent the cell abundance. However, as shown below, users can only input one gene each time, which is not a comprehensive approach to estimating cell lineage presence with many genes.

[3] Receptor ligand interaction

[3.1] Despite showing that there are 3 tumor subclones based on CNV clustering, the authors still only show interaction for “malignant”. The authors should redo their downstream analysis using the 3 distinct tumor subclones.

Response: Following the reviewer’s suggestion, we have performed the colocalization analysis using the updated cancer cell states explained in response to comment 1.1 of this reviewer. However, since we have not found any convincing interactions between either of two cancer states with other immune cell types, we only analyzed the cell-cell interactions between immune lineages in our manuscript.

Supplementary Fig. 15a, Colocalization analysis between cell types as Spearman correlations (Rho). The size of a node refers to the product of average fractions of a cell-type pair across all spots.

[3.2] The co-localization correlation-based approach seems to be similar to the approach described in the Stereoscope paper. The authors should compare and acknowledge this, or make differences in the approach more clear.

Response: Following the reviewer’s suggestion, we acknowledged the similarity with existing methods and compared the difference in the revised manuscript. SpaCE includes ligand-receptor (L-R) interactions beyond cell colocalization in the Stereoscope. The rationale for adding L-R interactions is that cell colocalization does not directly indicate physical interaction. The updated section in our manuscript is as follows:

“SpaCE explores cell-cell interaction by analyzing both cell colocalization and ligand-receptor (L-R) co-expression analysis across all ST spots. The former demonstrates co-occurrence of cell-type pairs whereas the latter provides evidence for cell-cell physical contacts by sending and receiving signals.

Cell colocalization was evaluated using the Spearman correlation between cell-type pairs across all spots. *We note that stereoscope also employed a similar strategy, but without considering L-R interactions within the same spots.*”

[3.3] The approach to interrogate potential receptor-ligand interaction (shuffling/permutation-based analysis) seems similar to CellPhoneDB and other tools which perform permutation to identify significant receptor-ligand interactions. The authors should compare and acknowledge this, or make differences in the approach more clear.

Response: We have cited the CellPhoneDB and compared the difference in the revised manuscript. CellPhoneDB randomly permutes the cell identities from scRNA-seq. However, such cell identity permutation is not possible for ST data because we can not separate the expression profile of single cells within a spot for cell identity shuffle. Instead, SpaCE computes an overall level of L-R interactions within an ST spot. Then, SpaCE tests whether ST spots with the colocalization of two cell types have more substantial L-R network scores than spots dominated by only one cell type. The background model is based on shuffling the ligand-receptor networks preserving directed degrees. The updated section in our manuscript is as follows:

“The overall level of L-R interactions within a spot was evaluated by 2,558 L-R pairs collected from a previous study. These ligands and receptors were filtered within genes detected by the ST platform. We shuffled the L-R interaction network by using BiRewire package to generate 1,000 randomized networks while preserving directed degree distributions. For a spot, an L-R network score is defined as the sum of expression products between all L-R pairs, divided by the average random value from 1,000 randomized networks. P-values were calculated with the empirical null distribution generated from network scores of randomized L-R interactions. Our strategy is distinct from CellPhoneDB to interrogate L-R interaction for scRNA-seq analysis, which randomly permutes the cluster labels of all cells. This strategy is not applicable to ST analysis because the expression profile of each cell in ST data is unknown due to mixed transcriptomics signals within spots. Thus, the computed L-R network score at each ST spot from SpaCE indicates the overall intensity of L-R interactions at each spot, but not specific interactions between two cell types.”

[3.4] The authors should also demonstrate the added value of the spatial information by comparing their receptor-ligand interaction analysis to that of matched scRNAseq data. This should validate the interactions that are found by SpaCE, but also highlight potential false positive in the scRNAseq data (i.e. where an interaction was predicted, but the cells do not co-localize) and false positive in the SpaCE analysis (i.e. where SpaCE predicted interactions, but such interactions cannot be recapitulated in the scRNAseq analysis)

Response: The reviewer’s comment is a fantastic suggestion! The reviewer mentioned two types of false positives, i.e., spatial only and single-cell only. 1) L-R pairs with spatial proximity but not from different cell-types, thus they do not represent interactions between distinct cell types; and 2) L-R pairs identified from single-cell data but without spatial proximity, thus they do not represent physical contacts between cells. We have highlighted these putative false positives and most confident interactions in our revised figure (Supplementary Fig. 16).

Supplementary Fig. 16. Ligand-Receptor interactions mediating CAF-M2 interactions.

a, Spearman correlation Rho between ligand and receptor expression across CAF-M2 colocalized spots of breast cancer ST data.

b, The number of scRNA-seq datasets that an L-R pair was estimated to be significant between CAF and M2 macrophage clusters in our single-cell data collection (Supplementary Table 2). Because ST spot data contains the mixture transcriptome from a few cell types, thus cannot directly reveal the cell source of gene expression. We integrated scRNA-seq data to reveal the cell origin that expressed each ligand or receptor. The X-axis shows putative L-R pairs mediating CAF-M2 interaction and two types of false positives. Spatial-only false positive is the L-R interaction happening in spatial proximity but not in single-cell data analysis. These L-R interactions could happen between the same cell type but not between two cell types. Single-cell-only false positive is the L-R interaction derived from dissociated single-cell data analysis but not in the spatial proximity.

Minor points

[4] The authors should present performance metrics, e.g. RAM and CPU time requirements.

Response: We have computed the CPU and RAM usage of our SpaCE and other methods in the ST simulation analysis (Fig. 2f). SpaCE performance is among the high-effective algorithm groups.

Fig. 2f, Comparison of running time and memory consumptions. All tools were run against a simulated ST data set of 1200 spots with default parameters.

[5] The title should make it clear that SpaCE only works for cancer.

Response: The title in our initial submission is “*Estimation of cell lineage and cell-cell interactions in tumors from spatial transcriptomics data*”. We would like the user to realize that SpaCE has a specific design for analyzing tumor ST data. However, as we described earlier, SpaCE can still accept a matched scRNA-seq dataset as customized references, indicating that SpaCE can be applied to any ST datasets. We also emphasized this point in the discussion part.

[6] “Spatial Transcriptomics” (e.g. line 41) is an ambiguous term. It describes a specific assay, but also the general body of methods to profile spatially resolved gene expression. Please change this to another term, e.g. “Spatially resolved transcriptomics”, and only use “Spatial Transcriptomics” when referring to the specific assay.

Response: In the introduction part of our manuscript, we used “spatially resolved transcriptomics” to represent the general methods, including imaging-based and sequencing-based methods. Then we introduced “Spatial Transcriptomics” representing sequencing based methods, e.g., Old ST, Visium, and Slide-Seq.

[7] The authors should consider using the term CNA instead of CNV as this is becoming more common to cancer genomics.

Response: We have already fixed it.

[8] The formula in figure 1c could be removed without losing anything major

Response: We have already removed it.

[9] Correlation values should be described as very high, high, moderate, and low

Response: We present correlation coefficients as values to present the original data and avoid defining cutoffs for very high, high, moderate, and low.

[10] Some of the p-value calculations are inappropriate for such large data. The authors should instead make use of effect size calculation (e.g. Cohens D).

Response: We have added the Cohen's D effect size calculation in the revised Fig. 3c and Fig. 5f involving large-scale data comparison.

[11] Line 38 – “genomics” should be “transcriptomics”

Response: We have already changed it.

[12] Line 53-54 could be supported with a reference.

Response: We have already added it.

[13] Line 56-57 could be supported with a reference.

Response: We have already added it.

[14] Line 57-59. This is a strong statement, and untrue given the approach used by Tangram. Also, Stereoscope and SPOTlight “should” not be affected by cellular density if I understand their approaches correctly. The authors should investigate if their statement is indeed true or remove it.

Response: As we state in the response to comment 1.2.4, SpaCE keeps a unidentifiable component in the regression, which reflects 1) the fraction of cell types not included in our hierarchical reference and 2) noise caused by low tissue density. Tangram can adjust cell density if the user input additional image segmentation information, which might not be accurate as described in response to comment 2.3 of this reviewer. Other methods, e.g., stereoscope and SPOTlight, normalize overall fractions to 1. We also further clarify it in the manuscript as the following.

“Further, cellular density may vary significantly across tumor regions; thus, cell fractions generated by existing decomposition methods, which normalize overall fractions to 1 in each spot, are not comparable across ST spots.”

[15] Line 163. Did the authors indeed evaluate accuracy or correlation?

Response: We have specified the meaning of performance metrics in the revision. For the simulated ST data, both Pearson correlation r and root mean square error (RMSE) were calculated between predicted cell fraction and ground truth (e.g., Fig. 2c and Supplementary Fig. 2b). For the real ST data, area under the ROC curve (AUC) values were calculated for each annotated region from H&E image by pathologists via double-blind annotations (e.g., Fig. 3d-e and Supplementary Fig. 12).

[16] Line 209. The statement “effectiveness of our algorithmic designs” should be “effectiveness of our algorithmic design in a simulated setting”.

Response: We have already changed it.

[17] After a thorough comparison to existing tools, the authors should revise which parts of their workflow are novel. To my understanding, this is limited to the use of inferCNV to identify the malignant cell proportions.

Response: Following the reviewer’s suggestion, we summarized the novelties of our workflow compared to existing tools in the revision. Briefly, the novelty parts are follows:

1) predict malignant cell abundance based on cancer type-specific CNA patterns.

2) for the CNA-low tumor, predict malignant cell abundance using cancer type-specific expression pattern. Such a dictionary-based approach is also broadly applicable to malignant cell identification in scRNA-seq data analysis, particularly for chromosomal stable tumors where the conventional inferCNV package does not work.

3) hierarchical regression for major lineage and sub lineage to avoid collinearity. Although Cell2location and MuSiC have provided a similar hierarchical regression function by asking users to define cell hierarchies manually, these methods do not provide a built-in reference profile. SpaCE is the only framework providing a comprehensive hierarchical cell-type reference in the tumor microenvironment (i.e., 12 major lineages and 20 sublineages)

4) add unidentifiable components in regression to capture unknown cell types and adjust cell density variations. Some previous methods, such as EPIC, Stereoscope, RCTD, and Cell2location, have an unknown component in regression. However, the EPIC framework treated this part as malignant cell fractions since the default EPIC version does not have procedures for searching cancer cell populations. The rest tools renormalized the overall cell fraction to 1 after the regression, thus removing this part in the final output.

References

1. Barkley, D. *et al.* Recurrence of cancer cell states across diverse tumors and their interactions with the microenvironment. doi:10.1101/2021.12.20.473565.
2. What is Visium for FFPE? -Software -Spatial Gene Expression -Official 10x Genomics Support. <https://support.10xgenomics.com/spatial-gene-expression/software/pipelines/latest/what-is-visium-ffpe>.
3. Liu, J. *et al.* Collagen 1A1 (COL1A1) promotes metastasis of breast cancer and is a potential therapeutic target. *Discov. Med.* **25**, (2018).
4. Zhang, Z., Wang, Y., Zhang, J., Zhong, J. & Yang, R. COL1A1 promotes metastasis in colorectal cancer by regulating the WNT/PCP pathway. *Mol. Med. Rep.* **17**, (2018).
5. Aran, D. *et al.* Reference-based analysis of lung single-cell sequencing reveals a transitional profibrotic macrophage. *Nat. Immunol.* **20**, 163–172 (2019).

REVIEWER COMMENTS

Reviewer #1 (Remarks to the Author):

The authors have much improved the paper, and addressed my concerns. The method seems to be improved substantially, and the authors have added comprehensive benchmarking. All in all, I feel that the paper as written could be a very useful for the community.

I have a few more comments, which I think it would be good for the authors to comment on, although a complete revision to address may be beyond the scope of the paper.

1. In the estimation of malignant cell abundance, the procedure was described as follows:

“For a tumor ST dataset, the ST-specific malignant expression profile was achieved by averaging the expression profiles of the spots in the malignant cell clusters from step 2. Subsequently, the expression profile of each spot within a tumor ST dataset was correlated to the ST-specific malignant expression profile.”

It is commonly known that tumor cells are heterogeneous and multiple subclones may co-exist on the same slide. Would the above strategy, which is based on the mean profile, be affected by this subclonal heterogeneity?

2. The hierarchical constrained non-negative least squares method in Stage 2 of the method is similar to the hierarchical constrained non-negative least squares method used in MUSiC, although the specific way that new unknown cell types are treated may be new (other methods also have tricks to deal with this). It would be good for the novelty to be made clear. The novelty of SPACE seems to be its specific and careful treatment of tumor cells in stage 1, and the combined ligand-receptor and cell co-localization analysis in stage 3. Co-localization, per se, is already performed by some existing spatial data analysis pipelines, e.g. Giotto, also most spatial papers have such a component such as:

<https://www.sciencedirect.com/science/article/pii/S2405471221003410>

<https://www.nature.com/articles/s41467-021-26271-2>

<https://www.sciencedirect.com/science/article/pii/S0092867420308709>

I recognize that often we can have a synthesis of pre-existing ideas, to build a new pipeline that has new merits. I emphasize that, so far, there is yet a pipeline that works well for tumor data, and this paper seems to partially fill that gap. But, it would be good to have a more detailed discussion, perhaps in the Intro and Discussion, contrasting the work with existing methods and transparently laying out the new ideas that are added, to more comprehensively distinguish the methodological novelty of this work.

Reviewer #3 (Remarks to the Author):

The authors have successfully addressed my comments and substantially improved the manuscript.

Reviewer #4 (Remarks to the Author):

General statement

I have reviewed the resubmitted work by Ru & Huang et al. In their revised work, they present better comparison to existing deconvolution tools, and demonstrate performance on other in-situ capture methods (the original ST and slide-RNAseq). In the original submission, my major enthusiasm arose from the inclusion of inferred CNA profiles as part of the deconvolution strategy. While other reviewers have raised the issue of the performance of CNV inference from single-cell transcriptomics data, this has now been replaced with an adhoc association of transcriptomic signatures and CNAs that is not benchmarked, which in my opinion is less scientific than the original approach. My overarching concern that SPaCE offer relatively little additional functionality over existing analytical frameworks remains (therefore it is my opinion that it would be more advantageous to present SPaCE as a novel deconvolution based tool, rather than an incomplete spatial analysis framework with some novel elements). While they have done a good job in demonstrating the improved performance of SPaCE over other tools for deconvolution, they attempt no comparative analysis of spatial receptor-ligand interaction tools.

Specific comments

Major

1) The new association of transcriptome with CNA signatures is interesting, but essentially similar to tools such as inferCNV in that they infer the CNA landscape from the transcriptome. The implementation of this approach in SPaCE is not well explored or benchmarked in the manuscript. Examples of interesting questions: (i) how do the best matching CNA identified by SPaCE compare to aCGH or WGS profiling of the same samples? (ii) How do the matched CNA profiles compare to inferCNV? (iii) How similar does the CNA profile have to be as part of the deconvolution procedure, or is it sufficient to just assign the spots as aneuploidy?

2) The receptor-ligand interaction analysis lacks the same rigour as used in the cell type decomposition. While I still believe that the major novelty of this study is in the deconvolution approach, the authors present 3 results sections on cellular communication without comparison to other tools (thus making a not insignificant part of the manuscript). For example, deeper investigation can be performed at 2 levels how does the choice of deconvolution tool affect R-L analysis, and how the choice of R-L tools affect the results. While I appreciate that the receptor-ligand interaction analysis is an immediate downstream analysis that is very interesting, there are important aspects that are not addressed in the manuscript, for example: (i) how does the performance of the R-L interaction analysis implemented in SPaCE rely on prior deconvolution via SPaCE, as compared to deconvolution by e.g. RTCD. (ii) how does the R-L interaction analysis implemented in SPaCE compare to other tools? I strongly believe that point 2i should be addressed, but 2ii might be out of scope, but should be mentioned in the discussion.

3) What aspect of novelty in the SPaCE workflow results in improved performance? The authors postulate that this arises from the (i) algorithmic design, and consideration of (i) tumour CNAs and (ii) unassigned portions. However, in figure S13, the comparisons of SPaCE without CNAs yield a difference of ~ 0.01 AUC for all 3 cell classes, and the unknown component shows differences of up to 0.02 UAC (negligible for tumour, ~ 0.02 for stroma and ~ 0.01 for immune). So the performance of SPaCE is still better than alternative tools, even without consideration of the CNA profile and unassigned portions. The remaining part of the algorithm is based on constrained nonnegative linear regression. Are there other aspects of the SPaCE algorithm that could explain the increased performance?

Intermediate

4) The paper would benefit from the depiction of how the spatial gene expression patterns are associated with cancer CNA signatures using one of the 8 samples as a comparison (ideally a tumour sample for which both spatial and genomic CNA data is available)

5) Calculation of statistics. How were the p-values generated for e.g. Fig 3c? To me it seems that the results for SPaCE were compared to the combination of all other methods, which is an inappropriate

comparison. Normally, when comparing methods, one would compare a single method to another method (therefore generating a p-value for each comparison, and correcting for multiple testing). This is likely going to lead to SPaCE not being significantly better than other tools such as Spatial DWLS and Cell2Loc for major lineages.

6) I still find parts of the manuscript to have ambiguous usage of “spatial transcriptomics” when referring to the group of technologies as a whole, vs the specific “Spatial Transcriptomics” assay. One example of this is when the authors refer to both the general technology and specific method in the manuscript, e.g. line 203 and 213. Perhaps the authors could refer to “in-situ capture” to help disambiguate this.

7) Recently a Tumor Immune Cell Atlas was published: <https://pubmed.ncbi.nlm.nih.gov/34548323/>. It would be great if the authors had a chance to revisit their analysis in light of tumor immune cell signatures.

8) Figure S4G shows exciting detections of subclones, but these seem not to be validated or cross-checked in the paper.

Minor

9) Line 19: I think the use of “geographical” is inappropriate as spatial organisation of tissue is not usually referred to as geography. Perhaps “topographical” is a suitable alternative. The authors can freely ignore this if they prefer to use the term “geographical”.

10) Line 19/20. “.. in tumors” could be changed to “... of tumor tissues”

11) Line 20. “... each tumor ST spot”. This statement makes a jump from general spatial transcriptomics to very specific in-situ capture-based methods. Perhaps the authors could rephrase the opening sentence to focus of “in-situ capture” methods, so that it is more clear when they describe “spots” in the second sentence.

12) Line 28. The wording suggests they applied it to “H&E” data. I believe the authors meant to say that SPaCE was applied to in-situ capture data with H&E staining as a ground truth.

13) Line 40-42. The authors could add a sentence that introduces “in-situ capture”

14) Line 44. There is ambiguity over an “ST spot”. An “ST spot” should have a diameter of 100um, but a “Visium spot” has a diameter of 55um. See my comment (6).

15) Line 45. Do the authors have a citation for Visium spots containing “up to 10 cells”? If not, the authors should generalise this statement to “multiple cells”.

16) Line 48 and others. “Cell lineage decomposition” is not the standard terminology in the field (in my opinion). I would rather suggest “cell type mixture decomposition” or “cell type decomposition”.

17) Line 48. “Bulk tissues” should be something like “bulk transcriptome profiling”.

18) Line 49. The statement “cannot address” is rather strong and not entirely proven at that point in the manuscript. Perhaps this should be toned down?

19) Line 51-53. The authors claim that guided deconvolution is not appropriate in light of ITH. However, the authors do not consider that multiple gene expression profiles could be used as reference by other existing methods.

20) Line 80. This is contradictory – you say that no “reference profiles” are used, but in the same sentence you say “dictionary of CNAs and malignant transcriptome signatures” are used. SPaCE clearly uses that as a reference dataset. The authors should rephrase this statement.

21) Line 83: The statement that scRNAseq data from tumors might be unsuitable (“Even using a scRNA-seq dataset from the same cancer type may be inappropriate due to inter-tumor heterogeneity.”), also argues against the suitability of bulk RNAseq data as used by the authors.

22) Line 193: It would be fair to mention that SpatialDWLS and RCTD also achieved $AUC > 0.75$ for all cell classes (similar to how authors point out that Stereoscope and RCTD have similar ROC to SPaCE for Lymphocytes).

Figures

General – many of the figures have fine grey lines/boxes rendered in Acrobat Reader on Windows. E.g. the cell type legend of Fig 2.

2B – text is hard to read

2F – the cut in the time is inelegant. Perhaps scaling the y-axis to $\log(\text{time})$ solves this.

3A – “Ovarian” label is misplaced.

4D – cell type colours are hard to distinguish. Change to have similar colours per class?

Reviewer #1 (Remarks to the Author):

The authors have much improved the paper, and addressed my concerns. The method seems to be improved substantially, and the authors have added comprehensive benchmarking. All in all, I feel that the paper as written could be a very useful for the community.

Response: We sincerely thank the reviewer for the constructive suggestions, which have significantly improved our manuscript.

I have a few more comments, which I think it would be good for the authors to comment on, although a complete revision to address may be beyond the scope of the paper.

1. In the estimation of malignant cell abundance, the procedure was described as follows:

“For a tumor ST dataset, the ST-specific malignant expression profile was achieved by averaging the expression profiles of the spots in the malignant cell clusters from step 2. Subsequently, the expression profile of each spot within a tumor ST dataset was correlated to the ST-specific malignant expression profile.”

It is commonly known that tumor cells are heterogeneous and multiple subclones may co-exist on the same slide. Would the above strategy, which is based on the mean profile, be affected by this subclonal heterogeneity?

Response: We thank the reviewer for raising this critical point. We agree that a tumor tissue slide for ST profiling might contain multiple subclones with distinct expression profiles. The strategy in SpaCE has been evaluated with simulated (Fig. 2e) and real (Fig. 3f) ST data sets with double-blind histopathology annotations. Since SpaCE has consistently high performance with one malignant reference, we keep the current simplified scheme. However, we still understand the reviewer’s concern and discussed this potential limitation in the Discussion.

“Several limitations exist for SpaCE. First, when estimating the malignant cell fraction (Fig. 1b), the ST-specific malignant expression profile was generated by averaging the expression profiles of all identified malignant cell spots within tumor ST data. As such, the estimation accuracy of malignant cell fractions might decrease for tumor ST data containing extremely distinct cancer cell states.”

2. The hierarchical constrained non-negative least squares method in Stage 2 of the method is similar to the hierarchical constrained non-negative least squares method used in MUSiC, although the specific way that new unknown cell types are treated may be new (other methods also have tricks to deal with this). It would be good for the novelty to be made clear.

Response: We thank the reviewer for the suggestions. In the revision, we have summarized the novelty and similarity to previous methods in the Discussion.

“Our strategy of hierarchical decomposition constrains the negative effect of collinearity among closely related cell types within a sublineage of decomposition results (Supplementary Fig. 2e). A previous tool MuSiC¹⁶ also performs hierarchical deconvolution by asking users to define cell hierarchies manually based on the clustering results of scRNA-Seq data. SpaCE provides a comprehensive hierarchical cell-type reference in the tumor microenvironment. Moreover, SpaCE includes an unidentifiable component to address cell types missing from the reference and tissues that may have areas with low local cellular density, which could otherwise lead to errors in deconvolution (Supplementary Fig. 4c-d and Supplementary Fig. 13c).”

3. The novelty of SPACE seems to be its specific and careful treatment of tumor cells in stage 1, and the combined ligand-receptor and cell co-localization analysis in stage 3. Co-localization, per se, is already performed by some existing spatial data analysis pipelines, e.g. Giotto, also most spatial papers have such a component such as:

<https://www.sciencedirect.com/science/article/pii/S2405471221003410>

<https://www.nature.com/articles/s41467-021-26271-2>

<https://www.sciencedirect.com/science/article/pii/S0092867420308709>

Response: The reviewer mentioned three representative studies involving spatial colocalization analysis, which can be grouped into two classes. The 1st and 3rd studies only apply to the sub-cellular resolution spatial technologies (e.g., seqFISH+ and CODEX), which provide transcriptomics readout above the single-cell resolution. In the current study, we focus on the multiple cellular (i.e., each spot may cover multiple cells) ST data, for which these strategies do not work. We also clarify their difference in the Discussion part.

“The spatial heterogeneity in different tumor regions is driven by tumor evolution and intercellular interactions between cancer cells and immune/stromal cells⁴⁵. Although several strategies⁴⁶⁻⁴⁸ have been developed for exploring cell-cell communications in spatial technologies with single-cell or sub-cellular resolutions (e.g., seqFISH+ or CODEX), they are not applicable to multiple cellular (i.e., spot level) ST data due to the spatial resolution limitation. SpaCE can combine co-localization and ligand-receptor analysis to study intercellular interactions when each ST spot captures multiple cells.”

The 2nd study did the deconvolution for multiple cellular ST data using Stereoscope, which has been included in our comparisons. Then, they calculated the linear correlation between different cell-type fractions to evaluate the colocalization. However, cell colocalization does not directly indicate physical interaction. Besides colocalization analysis, SpaCE provides further evidence for cell-cell interactions by analyzing ligand-receptor (L-R) interactions within ST spots. Previously, we have already clarified this difference in the Methods part.

“Cell colocalization was evaluated using the Spearman correlation between cell-type pairs across all spots. To further evaluate the significance of cell colocalization, we calculated the Spearman correlation between their reference profiles to rule out high colocalization

due to similar reference profiles. Previous methods, such as stereoscope⁸, included the cell colocalization analyses, but without considering L-R interactions. ”

4. I recognize that often we can have a synthesis of pre-existing ideas, to build a new pipeline that has new merits. I emphasize that, so far, there is yet a pipeline that works well for tumor data, and this paper seems to partially fill that gap. But, it would be good to have a more detailed discussion, perhaps in the Intro and Discussion, contrasting the work with existing methods and transparently laying out the new ideas that are added, to more comprehensively distinguish the methodological novelty of this work.

Response: We really appreciate the reviewer for recognizing the added value of SpaCE working on tumor ST data, and we agree with the reviewer about the importance of clarifying the novelty of SpaCE. Following the reviewer's comments, the methodological difference between our work with existing methods has been described in the Introduction, Discussion, and Methods parts, many of which have been shown in the aforementioned responses to points 1-3.

Reviewer #3 (Remarks to the Author):

The authors have successfully addressed my comments and substantially improved the manuscript.

Response: We thank the reviewer for the thoughtful comments and for reviewing our revised manuscript.

Reviewer #4 (Remarks to the Author)

General statement

a) I have reviewed the resubmitted work by Ru & Huang et al. In their revised work, they present better comparison to existing deconvolution tools, and demonstrate performance on other in-situ capture methods (the original ST and slide-RNAseq).

Response: We sincerely thank the reviewer for the constructive comments, which greatly helped us improve our analyses and manuscript.

b) In the original submission, my major enthusiasm arose from the inclusion of inferred CNA profiles as part of the deconvolution strategy. While other reviewers have raised the issue of the performance of CNV inference from single-cell transcriptomics data, this has now been replaced with an adhoc association of transcriptomic signatures and CNAs that is not benchmarked, which in my opinion is less scientific than the original approach.

Response: We understand the reviewer's concern. Theoretically, accurate inference of CNA in the tumor ST data is the optimal strategy for analyzing tumor cell fraction. However, inferred CNA values from inferCNV are strongly influenced by the selection of reference (i.e., normal) cells, resulting in false positive CNA regions. The new strategy can avoid this limitation based on CNA dictionaries from large genomics cohorts. We have validated the reliability of the new strategy in both simulated and real ST datasets. Please see the response to Point 1 below.

c) My overarching concern that SPaCE offer relatively little additional functionality over existing analytical frameworks remains (therefore it is my opinion that it would be more advantageous to present SPaCE as a novel deconvolution based tool, rather than an incomplete spatial analysis framework with some novel elements).

Response: We agree with the reviewer that SpaCE is a deconvolution tool rather than a complete framework, thus have clarified this point in the revision by changing the title and related parts in the manuscript. The new title is "Estimation of cell lineages ~~and cell-cell interactions~~ in tumors from spatial transcriptomics data".

d) While they have done a good job in demonstrating the improved performance of SPaCE over other tools for deconvolution, they attempt no comparative analysis of spatial receptor-ligand interaction tools.

Response: We agree with the reviewer about the lack of systematic comparisons on spatial ligand-receptor analysis. Following the reviewer's suggestion, we have explored the influence of deconvolution results on ligand-receptor (L-R) interaction analysis. Please see the response to Point 2 below.

Specific comments

Major

1a) *The new association of transcriptome with CNA signatures is interesting, but essentially similar to tools such as inferCNV in that they infer the CNA landscape from the transcriptome.*

Response: The new CNA dictionary method and inferCNV are distinct strategies. The previous inferCNV-based method predicted CNA values for genomics regions and genes from each ST transcriptomics data and determined the malignant cell abundance in each ST spot by the inferred CNA intensities. In contrast, the new method used the cancer type-specific CNA (or expression for chromosomal-stable cancer) patterns from TCGA genomics data to identify malignant cells whose expression profiles correlated with the TCGA patterns (Supplementary Fig. 1a, b). Thus, the new method did not infer the CNA landscape from transcriptomics data but used patterns from TCGA genomics data.

The new method has three advantages compared to the inferCNV-based strategy:

1. It avoids the prediction of CNA values from inferCNV, which is affected by the selection of reference profiles. Inappropriate normal reference might result in false positive CNA regions. Please see the response to Point 1c) (ii) below.
2. By using the cancer type-specific dictionary of both CNA and expression patterns (Supplementary Fig. 1a, right panel), the new method will also be applicable for ST data with chromosome-stable tumors, which cannot be analyzed by the inferCNV.
3. For each ST data, the new method only costs several minutes rather than the several hours required by the inferCNV.

Supplementary Fig. 1. Inference of malignant cell fraction.

a, Three steps to infer malignant cell fraction without any reference, based on a dictionary of cancer type-specific gene patterns. 1) Clustering all spots from a tumor ST dataset by hierarchical clustering. 2) Determining the malignant cell clusters whose spots have significant correlations with cancer type-specific patterns, with pattern selection rules shown in the right panel. 3) Estimating malignant cell abundance across all spots. For a tumor ST dataset, the ST-specific malignant expression profile was computed as the average expression profile among spots from malignant cell clusters in step 2. Then, the expression profile of each spot within a tumor ST dataset was correlated to the ST-specific malignant profile to infer the malignant cell fraction.

b, Copy number alteration (CNA) pattern across various data sets for the same cancer type in cBioPortal database. Red, amplifications; blue, deletions.

1b) The implementation of this approach in SPaCE is not well explored or benchmarked in the manuscript.

Response: We evaluated the new method using simulated (revised Fig. 2e) and real (revised Fig. 3f) ST data sets with double-blind histopathology annotations. The results showed that SpaCE, utilizing the CNA dictionaries to infer malignant cell fractions, performs better than other approaches.

Fig. 2e, Performance comparison between SpaCE and previous methods based on simulated ST data. A dot represents a simulated ST dataset synthesized from a single scRNA-seq dataset ($n=10$). The y value of an ST dataset presents the median Pearson correlation r between predicted and known cell fractions across cell types. All tools used the leave-one-out signature in panel d. The difference between SpaCE and other tools was evaluated by paired two-sided Wilcoxon signed-rank test. A star indicates that SpaCE is significantly better than others (BH-adjusted p value < 0.05). Bar height denotes average value across simulated ST datasets; error bars denote standard errors.

Fig. 3f, Performance comparison among methods based on double-blind pathology annotation. Each dot represents a dataset. Y-axis presents the area under the ROC curve (AUC) value of cell fraction decompositions for each method. The sub-panels represent the results in distinct tumor regions and the last sub-panel considered data from all three region types together. In each sub-panel, the difference between SpaCE and other tools was evaluated by the Wilcoxon signed-rank test. A star indicates that SpaCE is significantly better than others (BH-adjusted p value < 0.05). Bar height denotes average value across ST datasets; error bars denote standard errors.

1c) Examples of interesting questions:

(i) how do the best matching CNA identified by SPaCE compare to aCGH or WGS profiling of the same samples?

Response: We thank the reviewer for raising this point. However, the current tumor ST data (enumerated in Supplementary Table 3) do not have aCGH or WGS data on the same samples.

(ii) How do the matched CNA profiles compare to inferCNV?

Response: The reviewer raised an important point about comparing the CNA pattern from our dictionary to the inferred CNA profiles from inferCNV. As shown in Supplementary Fig. 1b, CNA profiles from bulk genomic data (from Affy SNP6) present consensus patterns.

Supplementary Fig. 1b, Copy number alteration (CNA) pattern across various data sets for the same cancer type in cBioPortal database. Red, amplifications; blue, deletions.

However, Reviewer #3 mentioned in the previous round that the predicted CNA value from inferCNV might be affected by the reference profile selection. For example, macrophages usually have high expression of HLA genes on chromosome 6, leading to false chromosome 6 CNA. We confirmed this phenomenon in the ST data as follows:

In the previous inferCNV-based method, the top 5% spots with high scores of normal cell markers were selected as reference cells, whereas the rest 95% of spots are observed cells. As such, different tumor ST datasets might select various normal cells (e.g., stromal cells or macrophages) as references because of tissue composition differences. To show the influence of reference selection, we used a breast cancer ST dataset as the example and chose macrophages, stromal, and macrophages+stromal spots as reference (top heatmap in the following figure) and the rest spots as observation (bottom heatmap).

From the inferCNV results, we observed that all three groups have several copy number gain regions in chromosomes 1 and 8 (green box in the following figure), which is consistent with the breast cancer bulk data (Supplementary Fig. 1b shown above). However, in the group with macrophages as reference (left panel), some deletion regions appear in the p arm of chromosome 6 (yellow arrow, left panel), which are not observed in the bulk genomics data. Similarly, with stromal cells as the references, some amplifications appear in the p arm of chromosome 6 (yellow arrow, right panel). These results might derive from cell lineage differences rather than the true CNA event.

Review Fig. 1. The inferCNV results of breast cancer ST data based on three types of references, including macrophages, macrophages+stromal, and stromal cell spots.

To make a quantitative comparison, we selected all HLA genes in the p arm of chromosome 6 and calculated their average inferCNV score for each spot. We observed a significant copy number loss reported by inferCNV with macrophages as normal controls (two-sided Wilcoxon signed-rank test $p < 2.2e-16$). In contrast, inferCNV reports a significant copy number gain for the same spots with stromal cells as references. Given these, we conclude that the original inferCNV-based method has large variability depending on the reference profile selection in ST data. Therefore, we finally decided to use the new strategy in the latest version of SpaCE.

Review Fig. 2. Inferred CNA values of HLA genes from inferCNV results. Each dot represents a spot. The value of a dot is the average inferCNV scores of HLA genes for a spot.

(iii) How similar does the CNA profile have to be as part of the deconvolution procedure, or is it sufficient to just assign the spots as aneuploidy?

Response: We thank the reviewer for raising this point. In step 2 of malignant cell inference (Supplementary Fig. 1a shown at the beginning of Response to Point 1), SpaCE identifies the malignant cell spots based on the similarity with the CNA pattern dictionary (revised Methods):

- i) the average coefficient r values of the spots within a cluster is significantly greater than 0 (One-sided Wilcoxon Signed rank test, $p < 0.05$);*
- ii) the proportion of spots positively correlated to the cancer type-specific signature (Pearson's $r > 0$ and two-sided correlation test $p < 0.05$) within a cluster is more than the proportion of ones in the whole ST dataset.*

We do not directly assign these spots as aneuploidy or set their malignant fraction as 1. Instead, we generate an ST-specific malignant profile to predict the malignant fraction of spots. As such, we get continuous malignant fractions across spots instead of the binary status of aneuploidy.

2) The receptor-ligand interaction analysis lacks the same rigour as used in the cell type decomposition. While I still believe that the major novelty of this study is in the deconvolution approach, the authors present 3 results sections on cellular communication without comparison to other tools (thus making a not insignificant part of the manuscript). For example, deeper investigation can be performed at 2 levels how does the choice of deconvolution tool affect R-L analysis, and how the choice of R-L tools affect the results. While I appreciate that the receptor-ligand interaction analysis is an immediate downstream analysis that is very interesting, there are important aspects that are not addressed in the manuscript, for example:

(i) how does the performance of the R-L interaction analysis implemented in SPaCE rely on prior deconvolution via SPaCE, as compared to deconvolution by e.g. RTCD.

Response: We agree with the reviewer about the importance of evaluating the impact of deconvolution approaches on the cell-cell interaction results based on the L-R analysis. The SpaCE cell-cell interaction analysis includes two steps: (1) cell colocalization and (2) ligand-receptor coexpression analysis. In step 1, linear correlations of inferred cell fractions are calculated to evaluate cell-type colocalization. In step 2, each spot was assigned an L-R

network score based on the L-R network and ST data. Combining these two steps, we test whether the colocalization between a pair of cell types has any physical support as ligand-receptor interactions. Following the review comment, in step 1, we replaced the deconvolution results from SpaCE with the estimated fractions from other deconvolution tools to identify the cell-cell interaction pair in the breast cancer ST data analyzed in the main manuscript (new Supplementary Table 4).

	SpaCE	SpatialDWLS	cell2location	CIBERSORTx	tangram	RCTD	MuSiC	stereoscope	SPOTlight	SCDC	EPIC
CAF-M2	✓	✓	✓	✓	✓	✓	✓	✓	✓	✓	✓
CAF-Endothelial	X	✓	X	✓	✓	✓	✓	✓	✓	✓	✓

Supplementary Table 4. *Estimated cell-cell interactions based on the decomposed cell fractions from different methods.*

After filtering by L-R interaction criterion, all methods have identified the interaction between CAFs and M2 macrophages. However, the prediction of CAF-Endothelial is not consistent across methods. SpaCE gives a conservative result by only reporting the CAF-M2 interaction. Since we lack the ground truth of cell-cell interactions in the breast cancer ST data, we can not evaluate whether CAF-Endothelial interaction is true.

(ii) how does the R-L interaction analysis implemented in SPaCE compare to other tools? I strongly believe that point 2i should be addressed, but 2ii might be out of scope, but should be mentioned in the discussion.

Response: Here, we summarized alternative L-R analysis methods into two classes. The 1st class is based on scRNA-seq data sets to infer L-R pairs from sender to receiver cells. However, our strategy calculates an overall L-R network score for each ST spot as evidence for cell-cell interactions instead of inferring specific L-R pairs. We describe this difference in the revised Methods part.

“We shuffled the L-R interaction network by using BiRewire⁴⁹ package to generate 1,000 randomized networks while preserving directed degree distributions. For a spot, an L-R network score is defined as the sum of expression products between all L-R pairs, divided by the average random value from 1,000 randomized networks. Our strategy is distinct from CellPhoneDB⁵⁰ to interrogate L-R interaction for scRNA-seq analysis, which randomly permutes the cluster labels of all cells. This strategy does not apply to ST analysis because the expression profile of each cell in ST data is unknown due to mixed transcriptomics signals within spots. Thus, the computed L-R network score at each ST spot from SpaCE indicates the overall intensity of L-R interactions at each spot, but not specific interactions between two cell types.”

The 2nd class applies to the sub-cellular resolution ST technologies that report transcriptomics data at single-cell levels (e.g., seqFISH+). In the current study, we focus on the multiple cellular ST data based on in situ capturing, for which these strategies do not work due to the resolution limitation. We also clarify their difference in the Discussion part.

“The spatial heterogeneity in different tumor regions is driven by tumor evolution and intercellular interactions between cancer cells and immune/stromal cells⁴⁵. Although several strategies^{46–48} have been developed for exploring cell-cell communications in spatial technologies with single-cell or sub-cellular resolutions (e.g., seqFISH+ or CODEX), they are not applicable to multiple cellular (i.e., spot level) ST data due to the spatial resolution limitation. SpaCE can combine co-localization and ligand-receptor analysis to study intercellular interactions when each ST spot captures multiple cells.”

3a) What aspect of novelty in the SPaCE workflow results in improved performance? The authors postulate that this arises from the (i) algorithmic design, and consideration of (i) tumour CNAs and (ii) unassigned portions. However, in figure S13, the comparisons of SPaCE without CNAs yield a difference of ~0.01 AUC for all 3 cell classes, and the unknown component shows differences of up to 0.02 AUC (negligible for tumour, ~ 0.02 for stroma and ~0.01 for immune). So the performance of SPaCE is still better than alternative tools, even without consideration of the CNA profile and unassigned portions.

Response: We thank the reviewer for pointing out these potentially confusing points.

The reviewer mentioned that in Supplementary Fig. 13b the comparisons of SPaCE with CNA and expression signatures yield a difference of ~0.01 AUC for all 3 cell classes. We hope to clarify that the goal of this analysis is not to compare the performance between the two approaches. Instead, this figure shows that both the CNA and expression signature can perform well and are comparable in real ST data deconvolution. SpaCE will use CNA approaches first to infer cancer cell spots and the alternative expression signature in case a tumor is chromosomally stable with low CNA. The comparisons between the cancer cell inference approach of SpaCE against others were shown in Fig. 3f and Supplementary Fig. 12 using real ST data. We have further clarified these points in the revision as follows:

“For the malignant cell quantification (Fig. 1b and Supplementary Fig. 1a), SpaCE prepared a pattern dictionary of both CNA and tumor-normal differential expression for diverse tumor types. All tumor ST data in our collection utilized the CNA pattern for cancer cell quantification due to significant correlations between spatial transcriptomic profiles and cancer-specific CNA patterns. However, we still used these ST data as surrogates to evaluate the expression signatures prepared for chromosomal stable tumors. We found the expression signatures achieved comparable performance (Supplementary Fig. 13b), supporting the reliability of our expression-based procedure, which will start its role for CNA-low tumors”

Meanwhile, we agree with the reviewer that SpaCE with an unidentifiable component only performs slightly better than SpaCE without one in stroma regions, which often have low cell density (Supplementary Fig. 13c). Thus, the inclusion of an unidentifiable component is not the major reason for better performance of SpaCE. We still keep the unidentifiable component because this feature will prevent inaccurate estimation of cell fractions when an unknown

lineage does exist and adjust cellular density variations across ST spots (Supplementary Fig. 4c-d).

3b) The remaining part of the algorithm is based on constrained nonnegative linear regression. Are there other aspects of the SPaCE algorithm that could explain the increased performance?

Response: We thank the reviewer for raising this point for clarification. We validated the advantage of hierarchical regression in the simulation part, where the ground truth fractions for major and sub lineages were known. The simulation analysis demonstrated that SpaCE with a hierarchical regression significantly outperformed SpaCE without one on decomposing both major and sublineages (Supplementary Fig. 2e).

Supplementary Fig. 2e,
Deconvolution results of SpaCE with (red bar) and without (gray bar) hierarchical lineage. A dot represents a simulated ST dataset synthesized from a single scRNA-seq dataset (n=10). Each simulated ST dataset was decomposed by using a leave-one-out signature, which is the reference derived from all scRNA-seq datasets except the one left

out to synthesize the simulated ST data. The y-axis presents the median Pearson correlation r between predicted and known cell fractions across cell types. The difference of groups was evaluated by the two-sided Wilcoxon rank sum test. Bar height denotes average value across simulated ST datasets; error bars denote standard errors.

Intermediate

4) The paper would benefit from the depiction of how the spatial gene expression patterns are associated with cancer CNA signatures using one of the 8 samples as a comparison (ideally a tumour sample for which both spatial and genomic CNA data is available)

Response: we agree with the reviewer that analyzing multi-modal data for samples with both spatial transcriptomics and genomic CNA data would improve our study further. However, the current public ST data lack these matched CNA data for the same tumor sample. Seven of the eight ST datasets from our manuscript have transcriptomics data only. Although one slide-RNA-seq data has matched slide-DNA-seq data, its CNA resolution is only on the scale of 1-Mb genomic bin along the chromosomes³³, which cannot enable gene-level analyses.

5) Calculation of statistics. How were the p-values generated for e.g. Fig 3c? To me it seems that the results for SPaCE were compared to the combination of all other methods, which is an inappropriate comparison. Normally, when comparing methods, one would compare a single method to another method (therefore generating a p-value for each comparison, and correcting for multiple testing). This is likely going to lead to SPaCE not being significantly better than other tools such as Spatial DWLS and Cell2Loc for major lineages.

Response: The reviewer mentioned Fig. 3c related to the role of an unidentifiable component in adjusting cell density variations. However, we postulate the reviewer actually refers to the performance evaluation of deconvolution methods in simulation (Fig. 2e) and real data (Fig. 3f) according to the other part of this comment. To avoid missing any points, we would respond to both of them.

In Fig 3c, we compared the unidentifiable component fractions or UMI counts across spots in both high and low cellular density regions from a breast cancer ST data set. The statistical test is a two-sided Wilcoxon rank-sum test as described in the figure legend.

Fig. 3c, Unidentifiable component fractions (left) and UMI counts (right) across spots in both high and low cellular density regions. The group values were compared by calculating the Cohen's d effect size and two-sided Wilcoxon rank-sum test. UMI: unique molecular identifier.

For the performance evaluation of deconvolution methods in the simulation (Fig. 2e), we followed the suggestions from the reviewer and compared SpaCE to every other method by using paired two-sided Wilcoxon signed rank tests. Please see the following figure. A white star means SpaCE is significantly better than the other method (BH-adjusted p value < 0.05). For major lineages, although the mean value of pearson r from SpaCE is higher than other methods, they are not significant for SpatialDWLS and Cell2Loc. For sub-lineages, SpaCE significantly outperformed all other methods.

Fig. 2e, Performance comparison between SpaCE and previous methods. A dot represents a simulated ST dataset synthesized from a single scRNA-seq dataset (n=10). The y value of an ST dataset presents the median Pearson correlation r between predicted and known cell

fractions across cell types. All tools used the leave-one-out signature in panel d. The difference between SpaCE and other tools was evaluated by paired two-sided Wilcoxon signed-rank test. A star indicates that SpaCE is significantly better than others (BH-adjusted p value < 0.05). Bar height denotes average value across simulated ST datasets; error bars denote standard errors.

We also conducted the same statistical test in real ST data with double-blind histopathology annotations (Fig. 3f). SpaCE still outperformed other methods, in particular when all types of regions were considered together.

Fig. 3f, Performance comparison among methods. Each dot represents a dataset. Y-axis presents the area under the ROC curve (AUC) value of cell fraction decompositions for each method. The sub-panels represent the results in distinct tumor regions and the last sub-panel considered data from all three region types together. In each sub-panel, the difference between SpaCE and other tools was evaluated by the Wilcoxon signed-rank test. A star indicates that SpaCE is significantly better than others (BH-adjusted p value < 0.05). Bar height denotes average value across simulated ST datasets; error bars denote standard errors.

6) I still find parts of the manuscript to have ambiguous usage of “spatial transcriptomics” when referring to the group of technologies as a whole, vs the specific “Spatial Transcriptomics” assay. One example of this is when the authors refer to both the general technology and specific method in the manuscript, e.g. line 203 and 213. Perhaps the authors could refer to “in-situ capture” to help disambiguate this.

Response: We thank the reviewer for raising this point for clarification. In the revision, we used spatial transcriptomics (ST) to represent the group of technologies as a whole, and in-situ capture as a part of ST methods. We also defined “pre-Visium” as the original generation of Visium from Ståhl et al in the introduction.

“Recent years have seen the rapid development of spatial transcriptomics (ST) with a range of gene coverage from a few targets to genome-wide and various cellular resolutions from subcellular to multiple cells^{2,3}. As a key branch of ST methods, in situ capturing strategy based on positional molecular barcodes enables unbiased capture of the whole transcriptome within intact tissue³. Its representative techniques include Slide-Seq⁴, 10x Visium⁵, and the early in-situ capturing method from which Visium was developed⁶.”

Following the reviewer's suggestion, we also revised line 203~213 shown below.

“We also demonstrated that SpaCE is applicable to a broad set of *in situ capturing* data with higher and lower resolutions.

.....

We also collected a *pre-Visium (an early in-situ capturing method from which Visium was developed)* dataset with a spot diameter of 100 μ m from pancreatic ductal adenocarcinoma, consisting of 428 spots covering 14,574 genes.”

7) Recently a Tumor Immune Cell Atlas was published:

<https://pubmed.ncbi.nlm.nih.gov/34548323/>. It would be great if the authors had a chance to revisit their analysis in light of tumor immune cell signatures.

Response: We thank the reviewer for bringing up this recent tumor immune cell atlas. As its literal meaning, this Tumor Immune Cell Atlas (TICA) is only composed of immune cells in the tumor microenvironment, but not malignant cells and stromal cells (e.g. fibroblasts and endothelial cells) in SpaCE signatures. Thus, we combined TICA cell signatures with SpaCE's stromal signatures and malignant cell inference module.

We evaluated them in two of our collected real ST data, which have both macrophage and lymphocyte annotation regions. Based on these two breast cancer ST datasets, we did another two runs of SpaCE shown in the following figure. One (green bar) is to replace the immune cells in our SpaCE atlas with TICA. The other one (blue bar) is to directly decompose ST data by using TICA immune cells only, in which the deconvolution results only contain immune cells. The red bar shows the output of SpaCE with default settings.

Supplementary Fig. 13d, Deconvolution results of SpaCE and TICA atlas. The AUC values of three runs for predicting different cell types (column) across ST datasets (row).

Based on these results, we can conclude that (1) the deconvolution performance of immune cells from TICA and SpaCE atlas are comparable (please compare the red and green bar); (2) if

only including immune cell reference in the deconvolution, the performance is quite worse (please see the blue bar). We have presented this comparison in the Methods section.

8) *Figure S4G shows exciting detections of subclones, but these seem not to be validated or cross-checked in the paper.*

Response: Thank you for this comment. First, we would clarify that we explored the different cancer cell states rather than subclones. As transcriptomics profiles of cancer cells are determined by both cancer cell subclonal status (e.g., somatic variations) and interactions with the surrounding environment, the ST data alone will not reveal subclones. Thus, we defined the clusters among cancer cells as “cancer cell states” following one recent study⁵⁴. We also admit that this part lacks validation and we regarded this analysis as an exploratory section in the supplementary material.

Minor

9) *Line 19: I think the use of “geographical” is inappropriate as spatial organisation of tissue is not usually referred to as geography. Perhaps “topographical” is a suitable alternative. The authors can freely ignore this if they prefer to use the term “geographical”.*

10) *Line 19/20. “.. in tumors” could be changed to “... of tumor tissues”*

11) *Line 20. “... each tumor ST spot”. This statement makes a jump from general spatial transcriptomics to very specific in-situ capture-based methods. Perhaps the authors could rephrase the opening sentence to focus of “in-situ capture” methods, so that it is more clear when they describe “spots” in the second sentence.*

Response: We have revised these lines following the reviewer’s comment.

*“Spatial transcriptomics (ST) technology **through in situ capturing** has enabled **topographical** gene expression profiling **of tumor tissues**. However, each tumor ST spot contains diverse immune and malignant cells, with cell densities that vary significantly across tissue regions.”*

12) *Line 28. The wording suggests they applied it to “H&E” data. I believe the authors meant to say that SPaCE was applied to in-situ capture data with H&E staining as a ground truth.*

Response: We have already revised it.

*“SpaCE provides higher accuracy than existing methods based on both simulation and **real ST data with matched H&E images and double-blind annotations as ground truth.**”*

13) *Line 40-42. The authors could add a sentence that introduces “in-situ capture”*

14) *Line 44. There is ambiguity over an “ST spot”. An “ST spot” should have a diameter of 100um, but a “Visium spot” has a diameter of 55um. See my comment (6).*

15) *Line 45. Do the authors have a citation for Visium spots containing “up to 10 cells”? If not, the authors should generalise this statement to “multiple cells”.*

Response: By following the reviewer’s comments, we have already revised this part.

“Recent years have seen the rapid development of spatial transcriptomics (ST) with a range of gene coverage from a few targets to genome-wide and various cellular resolutions from subcellular to multiple cells^{2,3}. As a key branch of ST methods, *in situ capturing strategy based on positional molecular barcodes enables unbiased capture of the whole transcriptome within intact tissue*³. Its representative techniques include Slide-Seq⁴, 10X Visium⁵, and the early *in-situ capturing method from which Visium was developed*⁶. Specifically, the commercial Visium platform can profile mRNA levels in both fresh-frozen and formalin-fixed paraffin-embedded (FFPE) tissues, enabling their widespread application⁷. However, *the spatial spot of various capturing strategies with a 10~100 μm diameter might measure a mixture of signals from multiple cells of different lineages.*”

16) Line 48 and others. “Cell lineage decomposition” is not the standard terminology in the field (in my opinion). I would rather suggest “cell type mixture decomposition” or “cell type decomposition”.

17) Line 48. “Bulk tissues” should be something like “bulk transcriptome profiling”.

18) Line 49. The statement “cannot address” is rather strong and not entirely proven at that point in the manuscript. Perhaps this should be toned down?

Response: Thank you for these comments. We have already revised them.

Many methods exist for *cell type decomposition* in general ST data⁸⁻¹³ and *bulk transcriptome profiling*¹⁴⁻¹⁷. However, *it is challenging* for these methods and their underlying strategies to address the unique *issue* of tumor ST data.

19) Line 51-53. The authors claim that guided deconvolution is not appropriate in light of ITH. However, the authors do not consider that multiple gene expression profiles could be used as reference by other existing methods.

Response: We agree with the reviewer and have removed this sentence.

20) Line 80. This is contradictory – you say that no “reference profiles” are used, but in the same sentence you say “dictionary of CNAs and malignant transcriptome signatures” are used. SPaCE clearly uses that as a reference dataset. The authors should rephrase this statement.

Response: The phrase “without any reference profiles” was deleted within this sentence.

“First, SpaCE infers malignant cell fractions ~~without any reference profiles~~ based on a gene pattern dictionary of copy number alterations (CNA) and malignant transcriptome signatures across common tumor types (Fig. 1b and Supplementary Fig. 1a).”

21) Line 83: The statement that scRNAseq data from tumors might be unsuitable (“Even using a scRNA-seq dataset from the same cancer type may be inappropriate due to inter-tumor heterogeneity.”), also argues against the suitability of bulk RNAseq data as used by the authors.

Response: Thank you very much for pointing this out. We have removed this sentence.

22) Line 193: It would be fair to mention that SpatialDWLS and RCTD also achieved AUC > 0.75 for all cell classes (similar to how authors point out that Stereoscope and RCTD have similar ROC to SPaCE for Lymphocytes).

Response: We have already added it to our manuscript.

“In general, SpaCE yielded more accurate estimates across cell types than other methods (Fig. 3f and Supplementary Fig. 12). Also, SpatialDWLS and RCTD achieved high performance for all cell types.”

Figures

General – many of the figures have fine grey lines/boxes rendered in Acrobat Reader on Windows. E.g. the cell type legend of Fig 2.

Response: Thank you for this comment. We have already fixed it.

2B – text is hard to read

Response: We have increased the resolution of Fig 2B.

2F – the cut in the time is inelegant. Perhaps scaling the y-axis to log(time) solves this.

Response: We have already changed it to a log scale on the y-axis.

Fig. 2f, Comparison of running time and memory consumptions. All tools were run against a simulated ST data set of 1200 spots with default parameters.

3A – “Ovarian” label is misplaced.

Response: We have already fixed it.

4D – cell type colours are hard to distinguish. Change to have similar colours per class?

Response: We thank the reviewer for the suggestion. Previously, we presented all cell major and sub lineages in Fig. 4d, resulting in the difficulty in distinguishing cell-type colors. In the revision, we split the major and sub lineages in revised Fig. 4d and new Supplementary Fig. 10d, respectively.

Supplementary Fig. 10d. Spatial localization of cell sublineages. The cell type of a Slide-seq bead is defined by the most abundant cell type in this bead.

REVIEWERS' COMMENTS

Reviewer #4 (Remarks to the Author):

Dear Authors,

I have reviewed the resubmitted work by Ru and colleagues. The method presented is a novel take on deconvolution of spatial gene expression data from cancer tissues, compared thoroughly to existing methods which do not consider tumor cells as part of their deconvolution model, and will certainly be of use to the community.

While the resubmitted work has addressed my comments and is suitable for publication, there are a few things the authors could additionally consider.

Author considerations:

1) The name of the tool is SpaCE (Spatial Cellular Estimator). Do the authors want to change the name to include a cancer element to the name?

2) The authors also have prior results from when they used inferCNV. These could be added to the results as a comparison to their current method to demonstrate superiority/non-inferiority.

3) The use of the term "Pre-Visium" seems awkward. In contradiction to my previous criticisms, using "Spatial Transcriptomics" here is easier to understand and technically more accurate.

4) Is there a visualisation of the gene pattern CNA dictionary? I have seen the images in Fig S1c, but these look more like chromosome wide dictionaries since the chromosomes are annotated, and not genes.

5) Is there a visualisation of the gene pattern CNA dictionary vs gene expression of clusters?

(Potential) corrections:

6) "In situ capture" is inconsistent ("in-situ" vs "in situ"). Should this be italicised?

7) In supplementary tables, it looks like the "square root" sign ($\sqrt{}$) is used instead of a tick/check mark (\checkmark).

8) Figure S13a. Should "Genes each spot" be "Genes in each spot"?

I would consider the above points as minor and at the discretion of the authors, and would not need to see a revision.

Reviewer #4 (Remarks to the Author):

I have reviewed the resubmitted work by Ru and colleagues. The method presented is a novel take on deconvolution of spatial gene expression data from cancer tissues, compared thoroughly to existing methods which do not consider tumor cells as part of their deconvolution model, and will certainly be of use to the community. While the resubmitted work has addressed my comments and is suitable for publication, there are a few things the authors could additionally consider.

Author considerations:

1) The name of the tool is SpaCE (Spatial Cellular Estimator). Do the authors want to change the name to include a cancer element to the name?

Response: We thank the reviewer for raising this point. We have renamed our tool as SpaCET (Spatial Cellular Estimator for Tumors).

2) The authors also have prior results from when they used inferCNV. These could be added to the results as a comparison to their current method to demonstrate superiority/non-inferiority.

Response: Following the reviewer's suggestion, we have added this comparison in our manuscript (please see Supplementary Fig. 13d). We can observe that the dictionary-based method (i.e., the default setting in SpaCET) outperforms the inferCNV-based strategy. Moreover, the current dictionary-based method only costs several minutes rather than the several hours required by the inferCNV-based strategy.

Supplementary Fig. 13d, Deconvolution results of SpaCET with distinct malignant cell prediction methods. The red bar shows the output of SpaCET with default settings whereas the grey bar presents the results of the inferCNV-based strategy.

3) The use of the term “Pre-Visium” seems awkward. In contradiction to my previous criticisms, using “Spatial Transcriptomics” here is easier to understand and technically more accurate.

Response: Since “Pre-Visium” only shows up twice throughout the whole manuscript, we just describe it in a sentence without any acronyms in the latest version of manuscript.

4) Is there a visualisation of the gene pattern CNA dictionary? I have seen the images in Fig S1c, but these look more like chromosome wide dictionaries since the chromosomes are annotated, and not genes.

Response: We thank the reviewer for raising this point for clarification. These figures indeed show CNA values in gene level, which are sorted by genes’ position along chromosomes. We also further clarify it in the legend of Fig S1c.

5) Is there a visualisation of the gene pattern CNA dictionary vs gene expression of clusters?

Response: Following the reviewer’s suggestion, we have visualized the correlation between the gene pattern CNA dictionary vs gene expression of clusters (please see Supplementary Fig. 4~11). The following figure just shows one of them as an example.

Supplementary Fig. 7c. Pearson correlation between the cancer type-specific CNA signature and the gene expression of spots in both malignant and non-malignant clusters. Each dot represents a spot. Two groups were compared by calculating the Cohen's d effect size and two-sided Wilcoxon rank-sum test.

(Potential) corrections:

6) “In situ capture” is inconsistent (“in-situ” vs “in situ”). Should this be italicised?

Response: we have gone over our manuscript and unified them as “in situ capturing”. Just like several recent review papers (such as <https://www.nature.com/articles/s41576-022-00515-3>), we did not set them in italicized format.

7) In supplementary tables, it looks like the “square root” sign ($\sqrt{\quad}$) is used instead of a tick/check mark (\checkmark).

Response: We have already fixed it.

8) Figure S13a. Should “Genes each spot” be “Genes in each spot”?

Response: We have already changed it.

I would consider the above points as minor and at the discretion of the authors, and would not need to see a revision.